# Immunogenicity Prediction with Dual Attention Enables Vaccine Target Selection

**Song Li**[1,3] *    **Yang Tan**[2] *    **Song Ke**[3]    **Liang Hong**[1,3]    **Bingxin Zhou**[1] †

[1] School of Physics and Astronomy, Shanghai Jiao Tong University.
[2] School of Information Science and Engineering, East China University of Science and Technology.
[3] Shanghai Matwings Technology Co., Ltd.

## Abstract

Immunogenicity prediction is a central topic in reverse vaccinology for finding candidate vaccines that can trigger protective immune responses. Existing approaches typically rely on highly compressed features and simple model architectures, leading to limited prediction accuracy and poor generalizability. To address these challenges, we introduce VENUSVACCINE, a novel deep learning solution with a dual attention mechanism that integrates pre-trained latent vector representations of protein sequences and structures. We also compile the most comprehensive immunogenicity dataset to date, encompassing over 7000 antigen sequences, structures, and immunogenicity labels from bacteria, virus, and tumor. Extensive experiments demonstrate that VENUSVACCINE outperforms existing methods across a wide range of evaluation metrics. Furthermore, we establish a post-hoc validation protocol to assess the practical significance of deep learning models in tackling vaccine design challenges. Our work provides an effective tool for vaccine design and sets valuable benchmarks for future research. The implementation is at `https://github.com/songleee/VenusVaccine`.

## 1 Introduction

Immunogenicity prediction is a pivotal step in reverse vaccinology. It aims at identifying specific proteins or peptide segments (protective antigens) that can produce humoral and/or cell-mediated immune responses and lead to memory cells production in the host organism (Adu-Bobie et al., 2003; Doneva et al., 2021). Protective antigens of pathogenic origin are potential vaccine candidates (PVCs) (Arnon, 2011). Rapid and accurate prediction of protective antigens can reduce research and development costs, minimize vaccine development risks, and provide safe and effective prevention strategies against emerging and reemerging infectious diseases that pose ongoing threats (Pizza & M., 2000; Bagnoli et al., 2011; Carvalho et al., 2021).

Current machine learning approaches commonly exploit the physicochemical properties of amino acid residues, as characterized by E-descriptors (Venkatarajan & Braun, 2001) or Z-descriptors (Hellberg et al., 1987), and transform these descriptors into a uniform vector representation using auto- and cross-covariance (ACC) methods. This vectorized format serves as input for machine learning models, which are then trained to predict whether antigens are protective or non-protective. (Doytchinova et al., 2007; Rawal et al., 2022). Due to the limited quantity of labeled data and the high complexity of the prediction problem, these methods excessively compress protein information in order to fit a relatively simple model, making it difficult to capture the complex relationship between antigens and their immunogenicity. As a result, these models often exhibit **limited prediction accuracy** and **inadequate generalizability** (*e.g.*, in cross-species immunogenicity predictions).

This study proposes VENUSVACCINE, a deep learning method that interprets immunogenicity based on the multimodal encoding of antigens, including their sequences, structures, and physicochemical properties. To supplement the inherent amino acid (AA) sequence information, two levels of structural tokens are employed to comprehensively capture antigen construction patterns, and the physicochemical properties are hand-crafted based on Z-descriptor and E-descriptor. When allocating protein representations from different modalities, we establish a *dual-attention mechanism*

---

*Equal contribution first authors.    † Corresponding author (bingxin.zhou@sjtu.edu.cn).

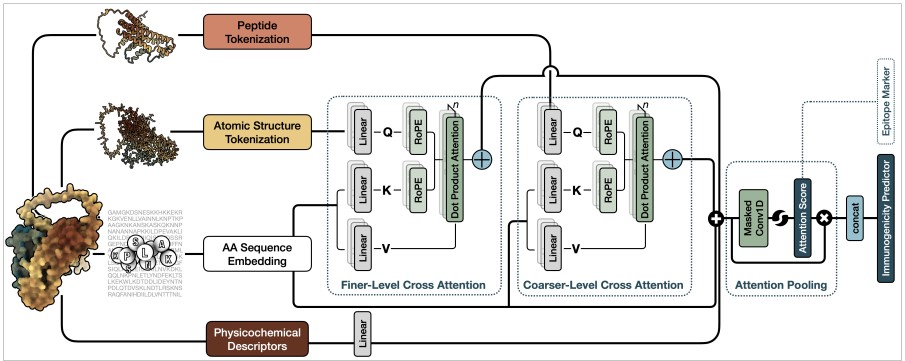

Figure 1: Illustrative framework of VENUSVACCINE. The model encodes sequence and structural representations using a dual-attention mechanism, followed by aggregation layers to incorporate global physicochemical attributes and perform binary classification of immunogenicity prediction.

(Figure 1) to facilitate communication between neighboring AAs at different scales. The use of tokenized structures at the atomic level (Van Kempen et al., 2023) and peptide level (Hayes et al., 2024) aligns with two biological intuitions: protein binding affinity primarily depends on atomic-level interactions, and local protein structures determine their functionality. Subsequently, we apply attention pooling to AA-level hidden representations to integrate them into protein-level vector representations, which incorporate the hand-crafted features that are commonly used to characterize the immunogenicity of proteins to further enhance the model's capacity for immunogenicity prediction.

The reliability of deep learning models requires thorough empirical validation, such as statistically significant tests on high-quality datasets and comparisons with similar baseline methods. While wet-lab experiments are considered the most reliable validation due to the inherent noise of high-throughput labels, the significant time and financial costs make it impractical for most research works to implement. For this purpose, we establish two types of evaluation protocols. First, we compile an immunogenicity database (**ImmunoDB**) with $7,216$ labeled antigens from bacteria, virus, and tumor. This benchmark enables large-scale validation of prediction methods and provides valuable training data for developing new models. Secondly, we propose two post-hoc validation protocols involving significant subjects (Helicobacter pylori and SARS-CoV-2), comprehensive evaluation datasets, and biologically meaningful evaluation criteria.

We verify the effectiveness of VENUSVACCINE using the two established types of validation approaches and provide extensive analysis. Based on **ImmunoDB**, we designed corresponding variations of training and test datasets, and performed comprehensive comparisons with existing methods across various evaluation metrics, focusing on prediction accuracy and generalizability of the trained model. The results demonstrate significant superiority of the proposed VENUSVACCINE from different perspectives. In post-hoc validations, we emphasize the practical significance of VENUSVACCINE by demonstrating that its assigned probability to protective antigens effectively helps users identify potential vaccine candidates for vaccine design.

In summary, this study addresses the key challenges of immunogenicity prediction in vaccine development by (i) introducing VENUSVACCINE, a deep learning supervised learning scheme with a dual attention mechanism; (ii) presenting the most comprehensive dataset of labeled antigens from different sources for training and testing immunogenicity prediction models; and (iii) constructing assessment protocols for evaluating prediction performance using benchmark datasets and post-hoc analyses. Empirically, we provide extensive evidence to validate that VENUSVACCINE meets the challenges of vaccine target selection by providing accurate and robust predictions. Moreover, the key AAs identified by VENUSVACCINE correlate with antigenic epitopes, suggesting more potential of our model to assist in vaccine development. Our work not only offers a reliable tool for advancing vaccine design but also provides crucial resources and insights for further research.

## 2 IMMUNOGENICITY PREDICTION AND PROTEIN LANGUAGE MODELS

**Problem Formulation**  Reverse vaccinology is an extensively utilized methodology for identifying prospective vaccine candidates by computationally screening the proteome of a pathogen (Dalsass

Figure 2: Data collection, redundancy processing, and dataset construction steps of **ImmunoDB**.

et al., 2019). This research is concerned with a binary classification task aimed at predicting the immunogenicity of proteins from three pathogenic sources: bacteria, virus, and tumor. The primary goal is to develop a model proficient in discerning whether a protein or peptide segment is a protective antigen or a non-protective antigen. Similar to most protein property prediction tasks, immunogenicity prediction also suffers from the limitation of scarce labeled data, making it crucial to develop a powerful and generalizable model. To achieve this goal, a feasible solution is to adopt a two-step training approach, where pre-trained models are first used to extract protein embeddings, followed by training the output module specifically for the prediction task.

**Pre-trained Protein Language Model for AA-level Encoding** Protein language models (PLMs) have become the dominant method for learning representations of protein sequences (Zhou et al., 2024a). The two primary pre-training strategies are masked language model (Meier et al., 2021; Rao et al., 2021) and autoregressive generation (Ferruz et al., 2022; Madani et al., 2023). The former involves predicting masked tokens in the input sequence while capturing co-evolutionary relationships between AAs within the same sequence. In contrast, the latter learns to iteratively generate the next token based on previous tokens, which is particularly useful for tasks like sequence design. As we focus on property prediction tasks, we adopt the first strategy. During training, random tokens in the input sequence are masked, and the objective is to optimize the model's parameters, denoted as $\boldsymbol{\theta}$, to minimize the error between the predicted and actual AAs at the masked positions, *i.e.*,

$$\arg\min_{\boldsymbol{\theta}} \mathbb{E}_{\boldsymbol{x} \sim \boldsymbol{X}} \mathbb{E}_M - \sum_{i \in M} \log \mathrm{P}(\boldsymbol{x}_i | \boldsymbol{x}_{/M}; \boldsymbol{\theta}). \tag{1}$$

The conditional probability $\mathrm{P}(\boldsymbol{x}_i | \boldsymbol{x}_{/M})$ for the $i$th token $\boldsymbol{x}_i$ in the sequence is determined by the unmasked tokens $\boldsymbol{x}_{/M}$. The model learns to capture the interactions between AAs within the protein sequence. Notably, such a sequential representation extraction method is not only applicable to AA sequence tokenization but can also be extended to capture protein structural features. A straightforward approach to achieve this is by expanding the output tokens to include not only AA types but also discretized structural tokens (Su et al., 2023; Heinzinger et al., 2023; Li et al., 2024).

## 3 IMMUNODB: NEW BENCHMARK FOR TRAINING AND EVALUATION

We curate three datasets with immunogenic (positive) and non-immunogenic (negative) labels from bacterial, viral, and human tumor sources. All positive samples and a part of negative samples are compiled through an exhaustive literature search with manual validation. We check publicly available sources, including PubMed, UniProt, NCBI, Protegen, IEDB, and other publications up to October 2023. The search is strictly limited to proteins proven to be immunogenic in humans. For all three datasets, the preparation steps include collecting positive samples, collecting negative samples, and filtering redundant and tail-region samples. An illustrative workflow is visualized in Figure 2. In total, we have $913/1562$ positive/negative instances in **Immuno-Bacteria**, $2078/1886$ in **Immuno-Virus**, and $300/477$ in **Immuno-Tumor**.

**Positive Samples** All positive samples are from the Protegen database and previous publications. For **Immuno-Bacteria**, data are from Dimitrov et al. (2020); Edison et al. (2020); Rawal et al. (2022). For **Immuno-Virus**, positive samples are gathered from Rawal et al. (2022); Doneva & Dimitrov (2024). For **Immuno-Tumor**, data are sourced from Sotirov & Dimitrov (2024). Notably, for **Immuno-Tumor**, both positive and negative samples are collected in this step. Protein or peptide sequences are from NCBI (Wheeler et al., 2007) and UniProtKB (Consortium, 2019).

**Negative Samples**    Due to the scarcity of data on non-immunogenic proteins in human studies, a part of the negative sets are collected from previous studies (Rawal et al., 2022; Sotirov & Dimitrov, 2024; Doneva & Dimitrov, 2024; Dimitrov et al., 2020; Li et al., 2023; Edison et al., 2020), and IEDB which the protein were validated by more than two B cell assays and identified as negative antigen in Homo sapiens, and others are curated following established methods reported in prior research (Dimitrov et al., 2020). For instance, when processing **Immuno-Bacteria**, we first download the proteomes of all of the bacteria involved from the UniprotKB, then conduct a BLAST search to identify sequences with less than 30% sequence identity to the protective antigens (positive samples). To further improve the possibility that the selected sequences are true-negative samples, we scored these sequences with VAXIJEN and kept the sequence where it is predicted to be `Probable Non-Antigen`. Finally, we exclude sequences shorter than 25 AAs or longer than 1024 AAs, and the remaining sequences are the negative samples for **Immuno-Bacteria**. A similar preprocessing approach is applied to the preparation of the negative dataset for **Immuno-Virus**. For **Immuno-Tumor**, we only perform the redundancy removal step, as the negative instances have already been provided by the literature and source database. Note that BLAST and VAXIJEN are used here to ensure that these selected sequences are as true-negative as possible, but this approach cannot be directly extrapolated to predict the immunogenicity of antigens, because it will predict the vast majority of sequences to be positive, which is a clear deviation from the empirical reality.

In summary, the new benchmark dataset for immunogenicity prediction is distinguished by its rigorous quality control, comprehensive data sourcing, and cross-species diversity. **ImmunoDB** undergoes a series of rigorous quality control measures, which include data cleaning and preprocessing steps. It helps in minimizing missing information, erroneous entries, and inconsistencies in data structure, which are critical for reducing noise and enhancing reliability. Meanwhile, **ImmunoDB** provides the largest and most comprehensive benchmark datasets in terms of size and species. This richness in data enhances the benchmark's applicability for developing and testing immunogenicity prediction models across various contexts. Moreover, including data from bacteria, viruses, and humans positions our dataset as a unique resource for cross-species analysis, thus supporting potential contributions to the robustness and generalizability of future models trained on them.

## 4    VENUSVACCINE: MODEL TRAINING AND INFERENCE

### 4.1    INPUT FEATURES

**Sequence Embedding and Structure Tokenization**    We use a pre-trained PLM to extract embeddings for protein sequences and structures. The *AA sequence embedding* can be extracted from, an encoder-only model (*e.g.*, ESM2; Lin et al. (2023)) or an encoder-decoder model (*e.g.*, ANKH; El-naggar et al. (2023)). For instance, given a protein sequence of length $L$, ESM2-650M delivers an $L \times 1280$ representation, with each AA being represented as a 1280-dimensional vector. For protein structures, we use two tokenizers to encode the backbone predicted by ESMFOLD (Lin et al., 2023). The first *atomic structure tokenization* uses FOLDSEEK (Van Kempen et al., 2023) to analyze the spatial relationships between atoms of neighboring AAs at a finer scale and summarize them into 20-dimensional tokens. The second *peptide tokenization* uses the ESM3 (Hayes et al., 2024) structure tokenizer to map coarser-scale substructures to $4,096$-dimensional structure tokens.

**Physicochemical Descriptors**    We incorporate five hand-crafted E-descriptors (Venkatarajan & Braun, 2001) and three Z-descriptors (Hellberg et al., 1987) to quantitatively characterize protein sequences. These descriptors provide an AA-level summary of physicochemical properties, such as hydrophobicity and secondary structure. They have been validated to assist in immunogenicity prediction (Doytchinova et al., 2007; Rawal et al., 2022). Details are in Appendix A.

### 4.2    DUAL ATTENTION

A protein's properties are determined by its sequence and structure. For immunogenicity prediction, focusing on local topological structures and molecular interactions may help identify exposed epitopes, thereby enhancing the extraction of critical information from the antigen. We combine hand-crafted features with intrinsic sequence and structural information derived from pre-trained models, as described above. These different categories of protein features are integrated using a dual attention mechanism, which guides the model to focus on important local regions of the proteins. The

features are then passed through attention pooling and propagated to the prediction head to obtain the predicted labels. The overall architecture of the model is shown in Figure 1. In the following sections, we detail the propagation rules of the dual attention and attention pooling modules.

**Finer-lever and Coarser-level Attention Module**  We construct the dual attention mechanism with two components, including *finer-level attention* and *coarser-level attention*. Both components construct a multi-head cross-attention framework to integrate *keys* and *values* from the sequence and *queries* from the structure. Following Lin et al. (2023), we replace the relative positional encoding of queries and keys with *Rotary Position Embeddings* (RoPE) (Su et al., 2024). For a protein sequence of $L$ AAs, *i.e.*, $\boldsymbol{x} = \{\boldsymbol{x}_1, \boldsymbol{x}_2, \ldots, \boldsymbol{x}_L\}$, we employ a pre-trained language model (*e.g.*, ESM2) to extract the sequence encoding $\boldsymbol{E}_{\text{seq}} \in \mathbb{R}^{L \times d}$. Structural encodings are extracted by structure tokenizers of different scales. We define the finer-level structural encoding from FOLDSEEK as $\boldsymbol{E}_{\text{fine}} \in \mathbb{R}^{L \times d}$ and the coarser-level structural encoding from ESM3 as $\boldsymbol{E}_{\text{coarse}} \in \mathbb{R}^{L \times d}$. Taking the finer-level attention as an example, the module's input is $\{\boldsymbol{E}_{\text{seq}}, \boldsymbol{E}_{\text{fine}}\}$, and the output is $\boldsymbol{H}_{\text{fine}}$. The queries $\boldsymbol{Q}$, keys $\boldsymbol{K}$, and values $\boldsymbol{V}$ for the attention layers are defined as follows:

$$\boldsymbol{Q} = \boldsymbol{E}_{\text{fine}}\boldsymbol{W}_Q, \quad \boldsymbol{K} = \boldsymbol{E}_{\text{seq}}\boldsymbol{W}_K, \quad \boldsymbol{V} = \boldsymbol{E}_{\text{seq}}\boldsymbol{W}_V, \tag{2}$$

where $\boldsymbol{W}_Q, \boldsymbol{W}_K, \boldsymbol{W}_V \in \mathbb{R}^{d \times d_k}$ are learnable projection matrices, and $d_k$ is the hidden dimension of queries and keys. Next, the positional information is attached to $\boldsymbol{Q}$ and $\boldsymbol{K}$ with RoPE, which captures relative positional relationships by applying a rotation to the embeddings based on their positions. The modified queries and keys are:

$$\tilde{\boldsymbol{Q}}_i = \text{RoPE}(\boldsymbol{Q}_i, i), \quad \tilde{\boldsymbol{K}}_j = \text{RoPE}(\boldsymbol{K}_j, j), \tag{3}$$

where $i$ and $j$ denote the positions in the respective sequences, and $\text{RoPE}(\cdot)$ is the RoPE function. Finally, the cross-attention $\text{Attn}(\cdot)$ is formulated as:

$$\text{Attn}(\boldsymbol{E}_{\text{seq}}, \boldsymbol{E}_{\text{fine}}) = \text{softmax}\left(\frac{\tilde{\boldsymbol{Q}}\tilde{\boldsymbol{K}}^\top}{\sqrt{d_k}}\right)\boldsymbol{V}, \tag{4}$$

where $\tilde{\boldsymbol{Q}} \in \mathbb{R}^{L \times d_k}$ and $\tilde{\boldsymbol{K}} \in \mathbb{R}^{L \times d_k}$. The coarser-level attention applies the same construction. The only difference is that the input queries are replaced with $\boldsymbol{E}_{\text{coarser}}$ from ESM3.

**Attention Pooling**  The processed AA-level features $\boldsymbol{H} = \text{concat}(\boldsymbol{E}_{\text{seq}}, \boldsymbol{H}_{\text{fine}}, \boldsymbol{H}_{\text{coarser}}, \boldsymbol{E}_{\text{ez}})$ are compressed using attention pooling and fed into the classifier for label prediction. Here, $\boldsymbol{H}$ for a protein consists of four components: the original sequence embedding, the joint sequence and structure representations processed by coarser-level and finer-level attention, and the handcrafted E and Z descriptors. Consider a processed feature sequence $\boldsymbol{H} \in \mathbb{R}^{L \times D}$, $L$ is the sequence length, and $D$ is the dimension of concatenated features. The attention pooling on $\boldsymbol{H}$ is defined as:

$$\text{AttnPool}(\boldsymbol{H}) = \sum_{i=1}^{L} \alpha_i \cdot \boldsymbol{H}_i, \quad \text{where } \alpha = \text{softmax}(\text{MaskedConv1D}(\boldsymbol{H})). \tag{5}$$

To obtain the final representation for the sequence, the model first computes $L \times 1$ attention scores using a one-dimensional masked convolution with a kernel size of 1 (Yang et al., 2023), then applies an input mask and passes it through a softmax function to obtain the attention weights $\alpha$. These attention weights not only guide the pooling process but also indicate the importance of each position for the final prediction. In subsequent analysis, we extract these weights as epitope markers. Relevant analysis can be found in the post-hoc analysis section of the Experiments.

**Prediction Head**  The output of the attention pooling $\text{AttnPool}(\boldsymbol{H})$ is sent directly to the final prediction head to produce the classification logits. This process involves projecting the pooled representation through a linear transformation, followed by a dropout layer, a ReLU activation layer, and a final linear layer: $\text{Logits}(\boldsymbol{H}) = \text{Linear}(\text{ReLU}(\text{DropOut}(\text{Linear}(\text{AttnPool}(\boldsymbol{H})))))$.

## 5 EXPERIMENTS

### 5.1 EXPERIMENTAL PROTOCOL

**Setup**  We built and trained the model according to the framework introduced in Section 4. For the sequence embedding, we consider three pre-trained PLMs, including ESM2 (Lin et al., 2023),

Table 1: Average performance of baseline methods in three immunogenicity prediction tasks from 10 random splits. The complete results with standard deviation can be found in Appendices.

| Model | Bacteria | | | | | Virus | | | | | Tumor | | | | |
|---|---|---|---|---|---|---|---|---|---|---|---|---|---|---|---|
| | ACC | T30 | MCC | F1 | KS | ACC | T30 | MCC | F1 | KS | ACC | T30 | MCC | F1 | KS |
| Random Forest | 81.1 | **77.7** | 57.1 | 69.7 | 0.60 | 88.3 | 98.7 | 76.7 | 88.5 | 0.80 | 66.8 | **63.5** | 27.0 | 46.4 | 0.42 |
| Gradient Boosting | 80.7 | 75.9 | 56.8 | 71.2 | 0.62 | 86.0 | 97.3 | 72.3 | 86.5 | 0.74 | 65.4 | 60.0 | 26.0 | 53.1 | 0.38 |
| XGBoost | 80.8 | 76.1 | 57.0 | 71.1 | 0.61 | 89.1 | **98.8** | 78.2 | 89.2 | 0.80 | 69.2 | 62.2 | 33.7 | 56.1 | 0.40 |
| SGD | 78.6 | 72.4 | 51.0 | 65.3 | 0.56 | 77.3 | 88.0 | 56.0 | 74.0 | 0.66 | 52.6 | 50.9 | 29.6 | 62.2 | 0.31 |
| Logistic Regression | 65.2 | 67.3 | 1.8 | 0.5 | 0.47 | 61.4 | 82.9 | 35.6 | 71.9 | 0.61 | 60.4 | 45.2 | 0.0 | 0.0 | 0.25 |
| MLP | 78.1 | 71.6 | 51.5 | 68.1 | 0.57 | 85.0 | 90.6 | 70.5 | 85.7 | 0.71 | 60.4 | 39.1 | 0.0 | 0.0 | 0.26 |
| SVM | 56.7 | 57.3 | 18.9 | 53.0 | 0.33 | 61.6 | 88.7 | 25.1 | 67.3 | 0.53 | 51.4 | 61.3 | 15.2 | 57.3 | 0.33 |
| KNN | 80.4 | 73.8 | 55.3 | 68.6 | 0.52 | 87.4 | 86.8 | 74.8 | 87.3 | 0.75 | 57.4 | 49.6 | 10.8 | 45.3 | 0.12 |
| Vaxi-DL | 68.1 | 55.5 | 41.7 | 66.9 | 0.42 | 65.3 | 62.2 | 33.4 | 72.9 | 0.29 | 54.9 | 41.3 | 8.5 | 47.7 | 0.10 |
| VaxiJen2.0 | 75.7 | 62.2 | 54.6 | 72.1 | 0.57 | 82.0 | 74.8 | 66.6 | 84.2 | 0.64 | - | - | - | - | - |
| VaxiJen3.0 | **83.3** | 58.7 | **63.2** | **75.1** | 0.63 | 68.1 | 62.5 | 42.3 | 75.4 | 0.35 | 39.0 | 35.7 | 0.0 | 55.9 | 0.00 |
| VirusImmu | - | - | - | - | - | 58.8 | 59.5 | 18.1 | 63.6 | 0.17 | - | - | - | - | - |
| VENUSVACCINE-Ankh | 82.3 | 78.5 | 60.3 | 73.3 | 0.64 | **92.2** | **99.7** | **84.3** | **92.3** | **0.85** | **76.9** | **73.5** | **55.0** | **73.5** | **0.61** |
| VENUSVACCINE-ESM2 | 80.6 | 77.0 | 57.0 | 71.7 | **0.66** | 90.3 | **99.5** | 80.6 | 90.6 | 0.82 | **74.0** | 61.7 | **46.1** | 68.2 | 0.58 |
| VENUSVACCINE-ProtBert | **84.5** | **84.5** | **65.9** | **77.0** | **0.66** | 91.4 | 97.4 | 82.8 | 91.2 | 0.84 | 71.5 | 68.7 | 44.7 | 67.6 | 0.54 |

† The top three are highlighted by **First**, **Second**, **Third**.

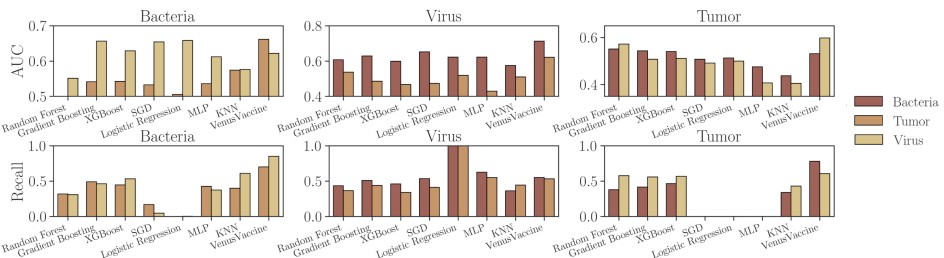

Figure 3: Generalizability of VENUSVACCINE by the AUC and Recall of cross-test evaluations. For instance, the top left figure reports the test performance on **Immuno**-Bacteria by different models trained on **Immuno**-Virus (yellow) and **Immuno**-Tumor (orange).

Ankh-large (Elnaggar et al., 2023), and ProtBert (Elnaggar et al., 2021). The source are listed in Appendix B. The model is optimized using ADAMW (Loshchilov et al., 2017) with a learning rate of 0.0005 and a weight decay of 0.01. In the attention pooling layer, dropout was set to 0.1. To ensure stable GPU memory usage during training, we limit the maximum number of tokens per batch to 4000, with a gradient accumulation of 1. The maximum training epoch was set to 50, with early stopping based on validation accuracy and patience of 5. All implementations were done using PyTorch (version 1.7.0), and experiments were run on an NVIDIA® RTX 3090 GPU with 24GB VRAM, mounted on a Linux server. The training process was tracked using WanDB. All source data and the implementation to reproduce the results will be publicly available upon acceptance.

**Baseline Methods** The performance of the proposed VENUSVACCINE is compared with existing machine learning methods and web server prediction tools. We select commonly used machine learning methods from the literature, including Random Forest, Gradient Boosting, XGBoost, SGD, Logistic Regression, MLP, SVM, and KNN, and follow the usual feature engineering approach by using ACC-transformed datasets to classify proteins as protective antigens (positive class) or non-protective antigens (negative class). We also include four popular web server methods specialized in antigen and vaccine candidate prediction: VAXIJEN2.0 (Doytchinova et al., 2007), VAXIJEN3.0 (Dimitrov et al., 2020), VAXI-DL (Rawal et al., 2022), and VIRUSIMMU (Li et al., 2023). VAXI-JEN2.0 employs an ACC transformation and PLS algorithm for antigen prediction. VAXIJEN3.0 extends it to include machine learning for broader biological entities. VAXI-DL utilizes deep learning models tailored for different disease-causing agents, and VIRUSIMMU applies an ensemble machine learning method with soft voting for viral immunogenicity prediction.

## 5.2 RESULTS ANALYSIS ON IMMUNODB

We evaluate the overall performance of the models on VENUSVACCINE. In light of the problems with existing methods, we focus on the accuracy and generalization of each method's predictive performance. The performance of the models is assessed using a range of metrics for a comprehensive evaluation, including AUC-ROC, accuracy, precision, recall, F1 score, MCC, Top K accuracy, cross-entropy, and KS statistics. Details are provided in Appendix C.

### 5.2.1 ACCURACY

We first investigate the overall performance of each model on the three datasets in **ImmunoDB**. For each dataset, we randomly split the data in a $7 : 1 : 2$ ratio and select the model with the highest accuracy on the $10\%$ validation set for evaluation. We repeat the assessment ten times for each test set, randomly selecting $50\%$ of the test data to calculate the scores for each metric, and report the average scores in Table 1. See Tables 4-6 of the Appendix for additional details.

All machine learning and PLM methods we retrained ensured non-overlapping among the training, validation, and test sets. However, the four web servers do not have open-source code available for retraining on each task, which may introduce some unfair advantages to them. Additionally, during data preparation, we follow the convention in the field by using VAXIJEN3.0 to check negative data labels, giving this method a natural advantage in predicting true negative data. Furthermore, since VAXI-DL is not available for predicting antigens from tumors, we did not test its performance on the **Immuno**-tumor dataset. Similarly, VIRUSIMMU is intended only for testing viruses, so we did not report its performance on Bacteria and Tumors for fairness consideration.

It is clearly demonstrated by the performance comparison of baseline methods across different metrics and datasets that VENUSVACCINE has a significant advantage in prediction accuracy (indicated by top ACC). It achieves high accuracy for both positive and negative samples (indicated by high MCC and F1 scores) and effectively distinguishes the distribution differences between positive and negative samples (indicated by high KS scores). Notably, we specifically report the positive rate of top predictions (T30), *i.e.*, the ratio of the top 30 samples with the highest predicted positive probabilities that are true positives. This metric is crucial in predictive tasks involving biological experiments since, in real-world scenarios, the samples are often highly imbalanced, and researchers usually conduct experimental validations on the samples with the highest model prediction confidence. Therefore, a model should provide high confidence for true positive samples. However, it is also important to note that if a model has a high T30 but considerably low accuracy, it may indicate that it occasionally predicts positive samples as negative, risking the omission of pivotal vaccine candidates. To avoid biased evaluations, we advocate assessing models based on their overall performance across multiple metrics rather than overly focusing on a single measure.

We also investigate the influence of protein structure quality on model prediction with an ablation study comparing structures predicted by AlphaFold2 and ESMFold (see Table 9 in Appendix). The performance metrics, detailed in the appendix, conclusively show that AlphaFold2's predicted structures lead to superior model performance, underscoring the importance of structural accuracy. While ESMFold's rapid prediction capability is noteworthy, AlphaFold2's higher precision in structure prediction significantly boosts model performance. This implies that when time is not a critical factor, it is advisable to opt for the more accurate structures provided by AlphaFold2. Unfortunately, the scarcity of available crystal structures limits our ability to contrast their performance with predicted ones, highlighting a need for more high-quality structural data in future studies.

### 5.2.2 GENERALIZABILITY

We next conduct cross-testing and compare the generalizability of different models (Figure 3). Here, the training and test sets are from different sources. The four web server-based methods are removed as we could not re-train them to avoid data leakage. We focus on AUC and recall to assess the model's overall learning ability and its capacity to predict true positives. Generally, models trained on the Virus dataset demonstrate higher generalizability compared to those trained on the Bacteria dataset, and those trained on Tumor performed the worst. This may be related to the dataset size, as the number of antigens in Virus is larger than in Bacteria, and both are larger than Tumor.

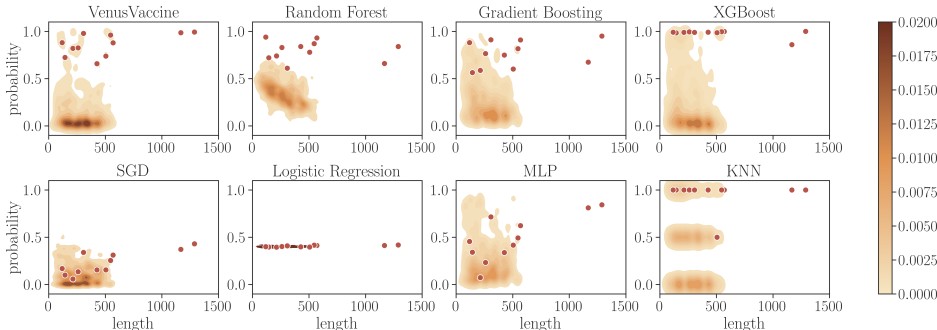

Figure 4: KDE of predicted immunogenicity scores on Helicobacter pylori candidates. The 11 experimentally determined immunogen are highlighted by red dots. Only VENUSVACCINE simultaneously identifies all positive samples while providing a reasonable overall distribution.

Moreover, excluding models with prediction issues (*e.g.*, logistic regression, where a 100% recall and a relatively small AUC suggest it trivially predicts all samples as negative), VENUSVACCINE consistently achieved top performance across the six test sets for both metrics. Also, the variation in VENUSVACCINE's performance on the same test set when trained on different datasets was relatively small, which further suggests the model's robustness in learning intrinsic patterns instead of overfitting to local variations in the data, thereby demonstrating stronger generalizabilities.

## 5.3 POST-HOC ANALYSIS

This section demonstrates the practical significance of VENUSVACCINE in addressing vaccine design challenges, as a supplement of the quantitative comparisons on well-established datasets and standard measurements. Two significant research subjects are selected to design corresponding ad-hoc analyses, including Helicobacter pylori and SARS-CoV-2. Specifically, in the first case, we examine *whether the potential vaccine candidates identified by* VENUSVACCINE *effectively cover immunogens*, which is frequently assessed by the enrichment of the model's predictions. In the second case study, we evaluate *whether the 'likely-effective' sequences identified by* VENUSVACCINE *are consistent with experimental evidence*, *i.e.*, if the model assigns the top confidence to the experimentally validated vaccine development sequences.

### 5.3.1 IDENTIFICATION OF POTENTIAL VACCINE CANDIDATES FOR HELICOBACTER PYLORI

In the first challenge, we use VENUSVACCINE to predict immunogens within the *Helicobacter pylori* proteome. Helicobacter pylori is a Group I carcinogen that can persist in the gastric mucosa, causing various gastrointestinal diseases (Malfertheiner et al., 2023). Identifying effective vaccine targets and developing corresponding vaccines can prevent Helicobacter pylori infections, thus significantly reducing the incidence of gastric diseases and lowering the rate of gastric cancer (Dos Santos Viana et al., 2021). To this end, we extract the proteome data from the **UniProtKB** database (ver 2024_04) (Consortium, 2019). A total number of $1,858$ sequences are extracted after redundancy removal. Although the majority of these $1,858$ sequences lack immunogenicity ground truth labels, we referr to the work of Dalsass et al. (2019), which record 11 of these sequences as protective antigens. We use the model trained on **Immuno**-**Bacteria** for inference since Helicobacter pylori is a bacterium. Note that there is no overlap between the training and prediction datasets.

When evaluate the prediction performance, two measurements are considered. (1) *Recall* ((9) in Appendix C) measures the fraction of true protective antigens identified within the set of PVCs predicted by VENUSVACCINE. It is a critical indicator of the model's ability to accurately retrieve known protective antigens from a larger set of candidates. (2) *Fold-Enrichment* (Dalsass et al., 2019) ((15) in Appendix C) measures the ratio of the true positive rate to the overall prevalence of the condition, characterizing the model's ability to enrich for positive samples.

To evaluate the performance of the model, we compare the results of VENUSVACCINE with other machine learning approaches. Table 16 compares the recall and enrichment of baseline methods, and Figure 6 visualizes the predicted kernel density estimation (KDE) of immunogenic probabilities

for candidates of varying sequence lengths, with the 11 immunogens highlighted in red dots. We exclude the VaxiJen series as they only provide web servers, in which case we could not confirm the absence of data leakage issues in the training data. Among the $1,858$ candidates, VENUSVACCINE identifies 123 PVCs. Notably, all the 11 true protective antigens are included within these 123 PVCs. A recall rate of $100\%$ demonstrates VENUSVACCINE's exceptional ability to extract known protective antigens from the proteome data. In comparison, Random Forest, Gradient Boosting, and XGBoost also achieve a $100\%$ recall rate, while other baseline models do not perform as well. Furthermore, VENUSVACCINE's fold-enrichment is 15.11, while the highest fold-enrichment among Random Forest, Gradient Boosting, and XGBoost is only 8.81. This indicates that VENUSVACCINE has a stronger predictive ability to enrich for positive samples, which is particularly beneficial when experimental testing is limited to a finite number of samples.

Figure 6 demonstrates three observations: (1) The methods in the second row have trouble correctly labeling all the protective antigens. (2) The overall probability distribution found by VENUSVACCINE aligns more closely with reality, with most samples clustering around probabilities close to 0, implying the majority of candidates are non-immunogenic. (3) VENUSVACCINE retains fewer PVCs from the candidates than baseline methods, allowing for the identification of the 11 protective antigens with fewer experimental efforts. The superior performance of VENUSVACCINE could be attributed to its effective integration of multimodal features by the dual-attention mechanism, which captures the complex interplay between sequence, structure, and physicochemical information. In summary, our model exhibits strong robustness and considerable potential to enhance the discovery of immunogenic vaccine targets in addressing the challenges posed by Helicobacter pylori.

### 5.3.2 UNRAVELING SARS-COV-2 PROTEOME FOR VACCINE TARGETING & EXPLANATION

In the second challenge, we aim to investigate if our VENUSVACCINE could effectively identify clinically validated vaccine targets from the SARS-CoV-2 proteome. The candidate test set consists of 38 sequences from the SARS-CoV-2 Data Hub (Wheeler et al., 2007). For the ground truth vaccine targets, we refer to the nine vaccines approved by the World Health Organization (Saravanan et al., 2024), five of which use the surface glycoprotein (spike protein) of the novel coronavirus as the vaccine target, while the other four use the whole virus, which also includes the spike protein. The overall prediction performance is in Table 17 in the Appendix, where VENUSVACCINE ranks the most important protein (spike protein) as the top among the 38 proteins. The model also identifies some new potential vaccine targets, *e.g.*, non-structural protein 9 (Nsp9).

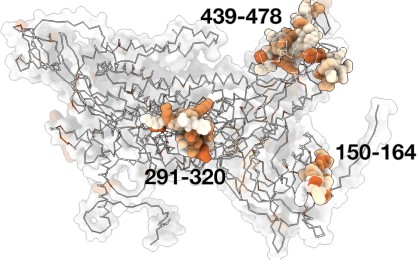

Figure 5: Epitope marker on the surface glycoprotein (NCBI ID: YP_009724390.1) by the attention score identifies vaccine targets.

To further validate and understand our predictions, we visualize the prediction results. Figure 5 maps the model's attention scores to the antigenic epitopes of the sequences and compares them with experimentally obtained epitopes (Shrock et al., 2020; Polyiam et al., 2021; Braun et al., 2024). It shows clearly that the model not only demonstrates excellent predictive performance but also possesses promising interpretability. For clear presentation, we display the chain trace of the spike protein and color the amino acids based on the attention scores from the attention pooling layer, with darker positions indicating higher attention scores. Overall, the model shows higher attention scores around position 400, and the spike protein contains three immunodominant linear B cell epitopes from position 319 to 541. Additionally, there are three regions with deeper colors that overlap with the SARS-CoV-2 spike receptor-binding domain (RBD), including positions 150-164, 291-320, and 439-478. This visualization provides a deeper understanding of the model's prediction process and enhances the credibility of our approach in identifying potential vaccine targets.

## 6 RELATED WORK

Developing machine learning models has become a popular approach for many predictive problems in vaccine design. One of the most widely used web servers for identifying potential vaccine can-

didates is VAXIJEN (Doytchinova et al., 2007), which employs an alignment-free method based on auto cross-covariance (ACC) transformation to identify antigens without relying on sequence alignment. Subsequent methods have introduced additional input features (He et al., 2010) or improved prediction models (Edison et al., 2020; Zhang et al., 2023; Rawal et al., 2022). Compared to other reverse vaccinology tools, these advanced techniques demonstrate superior performance in predicting bacterial protective antigens. Recently, Dimitrov *et al.* explored the immunogenicity of antigens from other sources, such as viruses (Doneva & Dimitrov, 2024) and tumors (Sotirov & Dimitrov, 2024), and evaluated various popular machine learning methods.

These machine learning approaches for predicting immunogenicity have often relied on limited datasets, typically confined to a specific type of pathogen, such as bacteria or viruses, and have been relatively small in scale. For instance, Doytchinova et al. (2007) trained and validated their models using only a few hundred samples. The scarcity of data not only impedes the development of robust predictive models but also restricts the ability to capture the complex immunological responses elicited by diverse antigens. To date, few studies have managed to amalgamate a comprehensive dataset that encompasses antigens from multiple pathogen classes, which is essential for training models capable of predicting immunogenicity with high fidelity across varied biological contexts.

Despite the rapid development of deep learning, the advancement of methodologies for protein property prediction, including immunogenicity prediction, has been relatively slow due to the limited amount of labeled data. Recently, BERT-style PLMs (Lin et al., 2023; Li et al., 2024) have gained increasing attention with an increasing number of prediction tasks fine-tuning models to better fit the specific targets (Schmirler et al., 2023; Tan et al., 2024b; 2025a), or attaching additional propagators to process sequence embeddings, such as geometric encoders (Yang et al., 2023; Tan et al., 2025b; Zhou et al., 2023; 2024b) or cross-modality aggregators (Tan et al., 2024a;c).

# 7 CONCLUSION AND DISCUSSION

Despite the rapid advancement of deep learning in providing novel solutions to numerous problems in scientific fields such as protein engineering, the application of advanced models remains largely limited to well-defined engineering problems with relatively mature and abundant data. However, many other issues with significant scientific and practical value have not received sufficient attention, and the methods used for these problems are considerably outdated. In addition to the oversight of important problems due to a lack of interdisciplinary cross-talk between research fields, another major reason is the absence of high-quality datasets and adequate evaluation methods for many specific issues. Preparing and processing datasets for model training and evaluation requires significant effort and domain knowledge, and validating the developed methods also requires feasible evaluation protocols that fit practical requirements. Ideally, these protocols should neither be as disconnected from practical needs as current computational metrics (*e.g.*, perplexity for sequence generation) nor as resource-intensive as wet lab validation processes in terms of time, money, and technical barriers.

As an exploratory work, this study takes immunogenicity prediction in vaccine development as an entry point to investigate how to prepare data, build models, and design evaluation strategies for a scientific problem that seeks powerful deep learning solutions. Although protein representation can be encoded by many pre-trained models, and immunogenicity prediction is inherently one supervised learning task, we emphasize that utilizing deep learning to solve scientific problems should not only focus on algorithm design but also on dataset construction and the practicality of evaluation methods. Here, we demonstrate the overall performance of the model using standard datasets and conventional quantitative evaluation methods, and analyze the model's reliability in specific applications using post-hoc analysis without conducting entirely new wet lab experiments.

ACKNOWLEDGMENTS

This work was supported by the grants from the National Natural Science Foundation of China (Grant Number 62302291; 12104295), the Computational Biology Key Program of Shanghai Science and Technology Commission (23JS1400600), Shanghai Jiao Tong University Scientific and Technological Innovation Funds (21X010200843), the Student Innovation Center at Shanghai Jiao Tong University, and Shanghai Artificial Intelligence Laboratory.

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
