## A    DETAILS ON PHYSICOCHEMICAL DESCRIPTORS

To provide protein-level properties, this study follows the convention and employs E-descriptors and Z-descriptors to quantitatively characterize protein sequences.

Venkatarajan & Braun (2001) proposed the E-descriptors in 2001. For each of the 20 naturally occurring amino acids, five numerical values are derived based on the principal component analysis (PCA) of 237 physicochemical properties. The first component E1 strongly correlates with the hydrophobicity of amino acids. E2 provides information about molecular size and steric properties. Components E3 and E5 describe the propensity of amino acids to occur in $\alpha$-helices and $\beta$-strands, respectively. E4 takes into account the partial specific volume, the number of codons, and the relative frequency of amino acids in proteins.

The Z-descriptors, defined by Hellberg et al. (1987), summarize the principal physicochemical properties of amino acids. These descriptors are derived by PCA of a data matrix consisting of 29 molecular descriptors such as molecular weight, pKas, NMR shifts, etc. The first component Z1 reflects the hydrophobicity of amino acids, Z2 their size, and Z3 their polarity. The E and Z descriptors have been widely used for the characterization and classification of proteins and are good predictors of immunogenicity (Doytchinova et al., 2007; Rawal et al., 2022).

Table 2: Description of E-descriptors and Z-descriptors.

| Name | Description |
|------|-------------|
| E1 | hydrophobicity of amino acids |
| E2 | molecular size and steric properties |
| E3 | the propensity of amino acids to occur in $\alpha$-helices |
| E4 | the partial specific volume, the number of codons, and the relative frequency of amino acids in proteins |
| E5 | the propensity of amino acids to occur in $\beta$-strands |
| Z1 | the hydrophobicity of amino acids |
| Z2 | size of amino acids |
| Z3 | polarity of amino acids |

## B    BASELINE METHODS

Here we list the sources for running the comparisons in the Experiment section. The machine learning methods are implemented using SKLEARN, the web server prediction tools are obtained from their official websites, and the pre-trained PLMs (for sequence embedding) are sourced from their respective open-source GitHub repositories. Table 3 provides details for accessing these methods.

Table 3: Source of baseline methods.

| Name | version | Source |
|------|---------|--------|
| VAXI-DL | - | https://vac.kamalrawal.in/vaxidl/ |
| VAXIJEN2.0 | - | http://www.ddg-pharmfac.net/vaxijen/VaxiJen/VaxiJen.html |
| VAXIJEN3.0 | - | https://www.ddg-pharmfac.net/vaxijen3/home/ |
| VIRUSIMMU | - | https://github.com/zhangjbig/VirusImmu |
| ESM2 | t33_650M_UR50D | https://huggingface.co/facebook/esm2_t33_650M_UR50D |
| Ankh | large | https://huggingface.co/ElnaggarLab/ankh-large |
| ProtBert | - | uniref&https://huggingface.co/Rostlab/prot_bert |

## C    EVALUATION METRICS

We employ a list of evaluation metrics to provide a comprehensive comparison of different prediction methods.

- AUC-ROC is a performance metric for classification models that measures the ability of the model to distinguish between classes. The ROC curve plots the True Positive Rate (TPR) against the

False Positive Rate (FPR) at various threshold settings. The AUC represents the area under this curve and provides an aggregate measure of performance across all thresholds.

$$\text{AUC} = P(\text{rank}(X_{\text{positive}}) > \text{rank}(X_{\text{negative}})) \tag{6}$$

- Accuracy measures the proportion of correct predictions made by the model out of all predictions.

$$\text{Accuracy} = \frac{\text{True Positives} + \text{True Negatives}}{\text{Total Predictions}} = \frac{TP + TN}{TP + TN + FP + FN} \tag{7}$$

- Precision quantifies the accuracy of positive predictions made by the model. It measures the proportion of true positives among all instances that were predicted as positive. High precision indicates that the model has a low rate of false positives.

$$\text{Precision} = \frac{\text{True Positives}}{\text{True Positives} + \text{False Positives}} = \frac{TP}{TP + FP} \tag{8}$$

- Recall also known as Sensitivity or True Positive Rate, quantifies the ability of the model to identify all relevant instances.

$$\text{Recall} = \frac{\text{True Positives}}{\text{True Positives} + \text{False Negatives}} = \frac{TP}{TP + FN} \tag{9}$$

- F1 score is the harmonic mean of Precision and Recall, providing a balance between the two metrics, especially when dealing with imbalanced datasets.

$$\text{F1 Score} = 2 \times \frac{\text{Precision} \times \text{Recall}}{\text{Precision} + \text{Recall}} \tag{10}$$

- MCC is a balanced measure for binary classifications, even if the classes are of different sizes. It considers true and false positives and negatives to produce a coefficient between -1 and +1.

$$\text{MCC} = \frac{(TP \times TN) - (FP \times FN)}{\sqrt{(TP + FP)(TP + FN)(TN + FP)(TN + FN)}} \tag{11}$$

- Top-K accuracy measures the fraction of times the correct label is among the model's top $K\%$ predictions.

$$\text{Top-K Accuracy} = \frac{\text{Number of correct predictions in top-K}}{\text{Number in top-K}} \tag{12}$$

- Cross-entropy quantifies the difference between two probability distributions: the true distribution and the predicted distribution by the model. $y$ is the true label (1 for positive class, 0 for negative class). $p$ is the predicted probability of the positive class.

$$\text{Cross-Entropy Loss} = -[y \log(p) + (1 - y) \log(1 - p)] \tag{13}$$

- KS statistics measures the maximum difference between the cumulative distribution functions (CDFs) of the positive and negative classes. It's used to evaluate the discriminatory power of a scoring model.

$$\text{KS} = \max_{x} |F_{\text{positive}}(x) - F_{\text{negative}}(x)| \tag{14}$$

- Fold-Enrichment measures the ratio of the true positive rate to the overall prevalence of the condition, characterizing the model's ability to enrich for positive samples.

$$\text{FE} = \frac{TP/(TP + NP)}{(TP + FN)/(TP + FP + TN + FN)} \tag{15}$$

## D    COMPLETE BASELINE COMPARISON ON IMMUNODB

Tables 4-6 present the detailed performance of all baseline methods, as well as our VENUSVAC-CINE, on the three benchmark datasets in **ImmunoDB**. The evaluation procedure follows the same methodology as described in the main text. We repeated the assessment on the test set for 10 times, with each time randomly selecting $50\%$ of the samples from the test set, and reported the average and standard deviation across these 10 runs in the tables. Due to limited space, in the main text we only report the average scores of the most important evaluation metrics. Here, we provide a more comprehensive summary, including additional measurements and confidence intervals, to allow for a more thorough comparison of the baseline models.

Table 4: Average performance of baseline methods in **bacteria** immunogenicity prediction tasks from 10 random splits, with the standard deviation reported after $\pm$.

| Model | AUC-ROC | Accuracy | Precision | Recall | F1 | MCC | TopK-30 | Cross Entropy | KS statistic |
|---|---|---|---|---|---|---|---|---|---|
| RF | **87.2±1.8** | 81.1±1.8 | 78.9±3.3 | 62.6±3.2 | 69.7±2.5 | 57.1±3.5 | 77.7±3.7 | 0.45±0.02 | 0.60±0.03 |
| GBT | 85.8±2.3 | 80.7±2.0 | 74.0±4.0 | 68.8±2.8 | 71.2±2.9 | 56.8±4.2 | 75.9±4.3 | 0.44±0.03 | 0.62±0.04 |
| XGBoost | 87.1±1.7 | 80.8±1.5 | 74.9±2.7 | 67.7±2.6 | 71.1±2.1 | 57.0±2.8 | 76.1±3.9 | 0.52±0.06 | 0.61±0.04 |
| SGD | 83.2±2.7 | 78.6±1.8 | 74.7±4.0 | 58.1±3.3 | 65.3±3.2 | 51.0±4.1 | 72.4±4.5 | 0.61±0.07 | 0.56±0.05 |
| Logit Reg | 77.7±3.0 | 65.2±2.6 | 20.0±4.0 | 0.2±0.5 | 0.5±0.9 | 1.8±3.5 | 67.3±5.9 | 0.64±0.01 | 0.47±0.04 |
| MLP | 82.4±2.9 | 78.1±1.5 | 69.0±3.9 | 67.3±2.9 | 68.1±2.8 | 51.5±3.5 | 71.6±5.0 | 0.49±0.03 | 0.57±0.05 |
| SVM | 68.1±2.5 | 56.7±2.1 | 42.6±3.7 | 70.3±3.7 | 53.0±3.6 | 18.9±4.4 | 57.3±4.8 | 0.64±0.01 | 0.33±0.04 |
| KNN | 82.5±2.4 | 80.4±1.7 | **77.2±4.4** | 61.9±3.6 | 68.6±3.1 | 55.3±4.1 | 73.8±5.8 | 3.95±0.69 | 0.52±0.04 |
| Vexi | 71.2±1.8 | 68.1±1.9 | 55.4±3.2 | 84.4±2.2 | 66.9±2.7 | 41.7±3.5 | 55.5±4.1 | 11.51±0.69 | 0.42±0.04 |
| VaxiJen2.0 | 78.5±1.5 | 75.7±1.7 | 61.0±2.3 | 88.3±1.8 | 72.1±1.9 | 54.6±2.8 | 62.2±3.0 | 8.76±0.61 | 0.57±0.03 |
| VaxiJen3.0 | 81.4±1.8 | **83.3±1.5** | 76.9±3.5 | **75.1±2.8** | **76.0±2.7** | **63.2±3.6** | 58.6±4.2 | 3.05±0.55 | 0.63±0.04 |
| VENUSVACCINE-Ankh | 86.6±2.1 | **82.3±2.0** | 76.8±5.0 | **70.4±3.2** | **73.3±3.0** | **60.3±4.6** | **78.5±4.5** | 0.97±0.10 | **0.64±0.03** |
| VENUSVACCINE-ESM2 | **87.6±1.9** | 80.6±2.0 | 73.3±3.1 | 70.2±4.0 | 71.7±3.3 | 57.0±4.6 | 77.0±4.4 | **0.46±0.05** | **0.66±0.03** |
| VENUSVACCINE-ProtBert | **88.7±1.1** | **84.5±1.5** | **83.8±1.6** | 71.3±3.0 | **77.0±2.2** | **65.9±3.0** | **84.5±1.6** | 0.51±0.04 | **0.66±0.02** |

† The top three are highlighted by First, Second, **Third**.

Table 5: Average performance of baseline methods in **virus** immunogenicity prediction tasks from 10 random splits, with the standard deviation reported after $\pm$.

| Model | AUC-ROC | Accuracy | Precision | Recall | F1 | MCC | TopK-30 | Cross Entropy | KS statistic |
|---|---|---|---|---|---|---|---|---|---|
| RF | 96.5±0.3 | 88.3±0.7 | 86.4±0.9 | 90.8±0.8 | 88.5±0.6 | 76.7±1.4 | 98.7±0.8 | **0.29±0.01** | 0.80±0.01 |
| GBT | 94.7±0.5 | 86.0±0.7 | 83.1±1.2 | 90.2±1.2 | 86.5±0.7 | 72.3±1.4 | 97.3±1.3 | **0.31±0.01** | 0.74±0.02 |
| XGBoost | 96.5±0.2 | 89.1±0.4 | 87.1±1.2 | 91.5±0.8 | 89.2±0.6 | 78.2±0.8 | **98.8±0.9** | **0.27±0.01** | 0.80±0.00 |
| SGD | 89.0±1.3 | 77.3±1.8 | 85.4±2.4 | 65.3±2.9 | 74.0±2.6 | 56.0±3.7 | 88.0±2.9 | 0.52±0.01 | 0.66±0.02 |
| Logit Reg | 85.7±1.4 | 61.4±1.0 | 56.3±1.4 | 99.6±0.2 | 71.9±1.1 | 35.6±1.5 | 82.9±2.8 | 0.68±0.00 | 0.61±0.02 |
| MLP | 91.2±1.0 | 85.0±1.2 | 81.2±1.7 | 90.8±1.6 | 85.7±1.3 | 70.5±2.3 | 90.6±2.5 | 0.38±0.02 | 0.71±0.02 |
| SVM | 76.8±1.8 | 61.6±1.2 | 58.3±1.4 | 79.6±2.1 | 67.3±1.1 | 25.1±3.1 | 88.7±1.9 | 0.71±0.01 | 0.53±0.03 |
| KNN | 90.6±1.0 | 87.4±0.8 | 87.4±1.5 | 87.2±0.9 | 87.3±0.8 | 74.8±1.6 | 86.8±2.3 | 2.91±0.33 | 0.75±0.02 |
| Vexi | 64.3±1.7 | 65.3±1.7 | 61.4±1.7 | 89.7±1.8 | 72.9±1.7 | 33.4±3.5 | 62.2±2.4 | 12.49±0.61 | 0.29±0.03 |
| VaxiJen2.0 | 82.1±1.0 | 82.0±1.2 | 75.1±1.8 | **95.7±1.0** | 84.2±1.3 | 66.6±2.1 | 74.8±3.4 | 6.47±0.44 | 0.64±0.02 |
| VaxiJen3.0 | 67.5±1.2 | 68.1±1.8 | 62.4±2.1 | 95.3±1.5 | 75.4±1.8 | 42.3±3.0 | 62.5±4.2 | 11.48±0.64 | 0.35±0.02 |
| VirusImmu | 58.7±2.2 | 58.8±2.1 | 56.8±2.5 | 72.2±2.1 | 63.6±2.1 | 18.1±4.4 | 59.5±4.6 | 14.86±0.76 | 0.17±0.04 |
| VENUSVACCINE-Ankh | **97.2±0.6** | **92.2±1.3** | **91.9±1.5** | 92.6±2.2 | **92.3±1.5** | **84.3±2.7** | **99.7±0.4** | 0.46±0.08 | **0.85±0.03** |
| VENUSVACCINE-ESM2 | 96.7±0.7 | 90.3±1.3 | **89.1±1.5** | 92.2±1.2 | 90.6±1.3 | 80.6±2.7 | **99.5±0.4** | 0.38±0.07 | **0.82±0.02** |
| VENUSVACCINE-ProtBert | 96.9±0.4 | 91.4±0.8 | **92.8±1.5** | 89.8±1.0 | **91.2±0.7** | 82.8±1.5 | 97.4±0.6 | 0.40±0.03 | **0.84±0.01** |

† The top three are highlighted by First, Second, **Third**.

Table 6: Average performance of baseline methods in **Tumor** immunogenicity prediction tasks from 10 random splits, with the standard deviation reported after $\pm$.

| Model | AUC-ROC | Accuracy | Precision | Recall | F1 | MCC | TopK-30 | Cross Entropy | KS statistic |
|---|---|---|---|---|---|---|---|---|---|
| RF | 74.4±5.3 | 66.8±4.2 | **63.6±7.9** | 37.0±8.4 | 46.4±8.3 | 27.0±9.7 | **63.5±5.2** | **0.59±0.03** | 0.42±0.09 |
| GBT | 71.3±5.1 | 65.4±3.2 | 57.0±4.4 | 49.8±6.5 | 53.1±5.3 | 26.0±6.9 | 60.0±5.8 | 0.62±0.06 | 0.38±0.08 |
| XGBoost | 74.2±4.5 | 69.2±3.6 | 64.1±5.3 | 50.2±7.5 | 56.1±6.4 | 33.7±7.8 | 62.2±5.5 | 0.68±0.08 | 0.40±0.08 |
| SGD | 65.6±3.4 | 52.6±3.0 | 45.5±2.8 | **98.7±1.6** | 62.2±2.5 | 29.6±3.8 | 50.9±7.8 | 0.77±0.02 | 0.31±0.06 |
| Logit Reg | 60.1±3.2 | 60.4±2.4 | 0.0±0.0 | 0.0±0.0 | 0.0±0.0 | 0.0±0.0 | 45.2±4.8 | 0.67±0.01 | 0.25±0.05 |
| MLP | 57.3±4.1 | 60.4±2.4 | 0.0±0.0 | 0.0±0.0 | 0.0±0.0 | 0.0±0.0 | 39.1±6.1 | 0.68±0.00 | 0.26±0.05 |
| SVM | 67.8±4.5 | 51.4±3.1 | 43.9±2.6 | 82.5±6.9 | 57.3±3.7 | 15.2±8.9 | 61.3±6.0 | 0.64±0.01 | 0.33±0.08 |
| KNN | 56.7±3.3 | 57.4±3.3 | 46.5±6.2 | 44.7±5.2 | 45.3±4.7 | 10.8±7.0 | 49.6±7.8 | 9.55±1.33 | 0.12±0.05 |
| VaxiJen2.0 | 54.3±4.3 | 54.9±4.2 | 44.7±6.5 | 51.4±4.6 | 47.7±5.3 | 8.5±8.6 | 41.3±9.2 | 16.27±1.53 | 0.10±0.07 |
| VaxiJen3.0 | 50.0±0.0 | 39.0±4.6 | 39.0±4.6 | **100.0±0.0** | 55.9±4.8 | 0.0±0.0 | 35.7±4.7 | 22.00±1.66 | 0.00±0.00 |
| VENUSVACCINE-Ankh | **85.0±2.7** | **76.9±3.1** | **65.2±4.1** | 84.6±1.6 | **73.5±2.7** | **55.0±5.2** | **73.5±3.6** | **0.48±0.05** | **0.61±0.05** |
| VENUSVACCINE-ESM2 | 80.7±4.5 | 74.0±4.3 | **66.1±6.9** | 70.6±6.3 | 68.2±6.4 | 46.1±9.2 | 61.7±9.3 | 1.37±0.28 | **0.58±0.06** |
| VENUSVACCINE-ProtBert | 79.5±4.2 | 71.5±4.1 | 59.3±4.8 | 78.9±6.7 | 67.6±4.6 | 44.7±8.3 | 68.7±8.4 | 0.56±0.05 | 0.54±0.09 |

† The top three are highlighted by First, Second, **Third**.

# E  ABLATION STUDY

This section discusses some of the design choices for our VENUSVACCINE. We evaluate the contributions of different inputs to the final results, as well as the advantages of attention pooling compared to simpler global pooling methods. Similar to the previous section, we conducted a full comparison across the three datasets, with results presented in Tables 12-14. The results suggest that encoding sequence information with PLMs is essential for enhanced performance in immunogenicity prediction tasks, and both atomic tokenization and physiochemical descriptors contribute significantly to

Table 7: Average performance of baseline methods in three immunogenicity prediction tasks with the 40% sequence identity cut-off between train and test set.

| Model | Bacteria | | | | | Virus | | | | | Tumor | | | | |
|---|---|---|---|---|---|---|---|---|---|---|---|---|---|---|---|
| | ACC | T30 | MCC | F1 | KS | ACC | T30 | MCC | F1 | KS | ACC | T30 | MCC | F1 | KS |
| Random Forest | 85.5 | **84.1** | 68.2 | 78.8 | 0.70 | 89.6 | 94.3 | 79.4 | 88.6 | 0.81 | 67.4 | 67.6 | 31.3 | 50.2 | 0.43 |
| Gradient Boosting | **86.4** | 82.7 | **70.3** | **80.8** | **0.74** | 78.3 | 91.4 | 55.6 | 77.0 | 0.74 | 66.3 | 65.2 | 29.3 | 55.6 | 0.41 |
| XGBoost | 82.0 | 82.7 | 61.4 | 75.5 | 0.66 | 90.8 | **97.1** | 80.7 | 89.4 | 0.86 | 70.7 | 69.0 | 38.8 | 59.5 | 0.43 |
| SGD | 72.2 | 74.6 | 38.0 | 41.7 | 0.59 | 80.0 | 92.9 | 61.4 | 75.4 | 0.79 | 55.1 | 48.6 | 30.9 | 64.7 | 0.32 |
| Logistic Regression | 63.5 | 63.6 | 0.0 | 0.0 | 0.45 | 59.2 | 82.9 | 31.2 | 68.9 | 0.73 | 58.0 | 48.1 | 0.0 | 0.0 | 0.24 |
| MLP | 78.5 | 75.0 | 53.5 | 70.3 | 0.62 | 88.8 | 92.9 | 76.8 | 86.9 | 0.83 | 58.0 | 45.2 | 0.0 | 0.0 | 0.27 |
| SVM | 54.9 | 61.4 | 20.7 | 56.2 | 0.38 | 77.9 | 84.3 | 53.7 | 75.4 | 0.68 | 55.4 | **70.5** | 20.5 | 61.0 | 0.40 |
| KNN | 84.3 | 79.1 | 65.6 | 77.2 | 0.64 | **91.7** | **97.1** | **83.4** | **90.6** | 0.84 | 59.6 | 50.5 | 16.1 | 49.6 | 0.18 |
| VENUSVACCINE-Ankh | **87.6** | **89.1** | **73.1** | **82.2** | **0.75** | 87.1 | 95.7 | 74.7 | 84.4 | **0.90** | **81.4** | **84.3** | **63.3** | **79.3** | **0.67** |
| VENUSVACCINE-ESM2 | 82.3 | 83.2 | 62.9 | 77.2 | 0.73 | **92.1** | **100.0** | **84.5** | **91.2** | **0.94** | 73.9 | 62.4 | **45.7** | 67.5 | **0.56** |
| VENUSVACCINE-ProtBert | **86.9** | **88.6** | **71.7** | **81.8** | **0.81** | **92.9** | **100.0** | **85.9** | **92.0** | **0.96** | 73.4 | **73.3** | 47.8 | **70.9** | **0.56** |

† The top three are highlighted by First, Second, **Third**.

Table 8: Average performance of VENUSVACCINE in **Bacteria** immunogenicity prediction tasks from 5 random data split with train/valid/test=8:1:1, with the standard deviation reported after $\pm$.

| Model | AUC-ROC | Accuracy | Precision | Recall | F1 | MCC | TopK-30 | Cross Entropy | KS statistic |
|---|---|---|---|---|---|---|---|---|---|
| VENUSVACCINE-Ankh | 87.3±2.2 | 81.3±1.9 | 80.1±5.6 | 70.8±8.2 | 74.7±2.8 | 60.7±3.0 | 84.9±1.8 | 0.85±0.24 | 0.63±0.03 |
| VENUSVACCINE-ESM2 | 87.4±1.5 | 80.3±1.3 | 77.7±3.4 | 69.8±3.6 | 73.4±2.2 | 58.1±2.9 | 81.4±3.4 | 0.63±0.30 | 0.62±0.04 |
| VENUSVACCINE-ProtBert | 88.4±1.8 | 81.9±1.8 | 77.8±2.6 | 74.8±6.2 | 76.1±2.4 | 61.7±3.9 | 83.8±3.2 | 0.50±0.10 | 0.66±0.05 |

the representation of the protein. The combination of all three modules achieves optimal prediction performance. The ablation experiments of different protein structure folding methods are shown in Tables 9-11

## F ADDITIONAL INFORMATION ON THE ASSESSING DATASET IN CASE STUDY

Tables 16-17 summarize the prediction results on Helicobacter pylori and SARS-COV-2 targets for the two case studies in the post-hoc analysis. The relevant analysis can be found in the main text.

Table 9: Ablation study on different folding methods on **ImmunoDB-Bacteria**.

| Fold Type | Model | Accuracy | TopK | MCC | F1 | KS-statistic |
|---|---|---|---|---|---|---|
| ESMFold | VENUSVACCINE-Ankh | 82.7 | 78.8 | 61.8 | 75.0 | 0.63 |
| | VENUSVACCINE-ESM2 | 80.4 | 77.0 | 56.4 | 70.3 | 0.61 |
| | VENUSVACCINE-ProtBert | 82.5 | 80.1 | 61.2 | 74.1 | 0.66 |
| AlphaFold2 | VENUSVACCINE-Ankh | 82.3 | 78.5 | 60.3 | 73.3 | 0.65 |
| | VENUSVACCINE-ESM2 | 80.7 | 77.0 | 57.0 | 71.7 | 0.66 |
| | VENUSVACCINE-ProtBert | 84.5 | 84.5 | 65.9 | 77.0 | 0.66 |

Table 10: Ablation study on different folding methods on **ImmunoDB-Virus**.

| Fold Type | Model | Accuracy | TopK | MCC | F1 | KS-statistic |
|---|---|---|---|---|---|---|
| ESMFold | VENUSVACCINE-Ankh | 91.5 | 97.5 | 83.0 | 91.4 | 0.85 |
| | VENUSVACCINE-ESM2 | 89.0 | 96.2 | 78.2 | 89.2 | 0.80 |
| | VENUSVACCINE-ProtBert | 89.7 | 98.7 | 79.6 | 89.4 | 0.83 |
| AlphaFold2 | VENUSVACCINE-Ankh | 92.2 | 99.7 | 84.3 | 92.3 | 0.85 |
| | VENUSVACCINE-ESM2 | 90.3 | 99.5 | 86.6 | 90.6 | 0.82 |
| | VENUSVACCINE-ProtBert | 91.4 | 97.4 | 82.8 | 91.2 | 0.84 |

Table 11: Ablation study on different folding methods on **ImmunoDB-Tumor**.

| Fold Type | Model | Accuracy | TopK | MCC | F1 | KS-statistic |
|---|---|---|---|---|---|---|
| ESMFold | VENUSVACCINE-Ankh | 77.7 | 77.4 | 55.8 | 74.5 | 0.62 |
| | VENUSVACCINE-ESM2 | 73.7 | 67.8 | 45.5 | 67.1 | 0.58 |
| | VENUSVACCINE-ProtBert | 70.3 | 67.8 | 41.6 | 67.1 | 0.51 |
| AlphaFold2 | VENUSVACCINE-Ankh | 76.9 | 73.5 | 55.0 | 73.6 | 0.61 |
| | VENUSVACCINE-ESM2 | 74.0 | 61.7 | 46.1 | 68.2 | 0.58 |
| | VENUSVACCINE-ProtBert | 71.5 | 68.7 | 44.8 | 67.6 | 0.54 |

Table 12: Average performance of baseline methods in **Bacteria** immunogenicity prediction tasks from 10 random splits, with the standard deviation reported after $\pm$.

| input module | AUC-ROC | Accuracy | Precision | Recall | F1 | MCC | TopK-30 | Cross Entropy | KS statistic |
|---|---|---|---|---|---|---|---|---|---|
| Sequence (One-hot) | 87.1±1.0 | 79.1±1.2 | 70.0±2.6 | 75.7±2.7 | 72.7±2.1 | 55.9±2.8 | 78.7±2.8 | 0.45±0.02 | 0.61 ±0.02 |
| Ankh | | | | | | | | | |
| Sequence (PLM) | 86.7±1.7 | 81.5±1.5 | 78.0±2.8 | 70.7±2.5 | 74.2±2.3 | 60.0±3.3 | 63.1±3.2 | **0.42±0.02** | 0.84±0.04 |
| Sequence (PLM)+Structure Token | 86.7±1.9 | 83.1±1.5 | 82.2±2.2 | 70.2±4.1 | 75.7±2.7 | 63.4±3.3 | 63.1±3.6 | 0.45±0.04 | **0.85±0.03** |
| Sequence (PLM)+Descriptors | 87.1±1.7 | 82.2±1.6 | 77.1±3.4 | 72.4±3.5 | 74.6±2.7 | 61.1±3.6 | 63.4±3.9 | 0.42±0.03 | 0.81±0.05 |
| VENUSVACCINE | **89.8±1.7** | **86.2±0.8** | **84.5±1.3** | **77.7±2.3** | **80.9±1.5** | **70.3±1.7** | **88.7±4.0** | 0.44±0.04 | 0.71±0.03 |
| ESM | | | | | | | | | |
| Sequence (PLM) | 88.8±1.4 | 83.0±1.5 | 78.4±2.4 | 72.3±2.8 | 75.2±2.4 | 62.4±3.3 | 68.7±3.0 | **0.39±0.03** | 0.80±0.03 |
| Sequence (PLM)+Structure Token | **89.9±1.4** | 84.0±1.0 | **81.0±2.3** | 74.8±3.0 | 77.8±2.0 | 65.4±2.3 | 68.8±2.9 | 0.41±0.03 | **0.85±0.03** |
| Sequence (PLM)+Descriptors | 88.2±1.1 | 82.0±1.5 | 76.5±1.7 | 75.2±3.4 | 75.8±1.8 | 61.5±2.8 | 67.1±2.8 | 0.41±0.02 | 0.80±0.03 |
| VENUSVACCINE | 89.8±1.1 | **85.0±0.5** | 79.1±2.3 | **80.8±1.4** | **80.0±1.1** | **67.9±1.3** | 85.2±3.4 | 0.45±0.03 | 0.70±0.01 |
| ProtBert | | | | | | | | | |
| Sequence (PLM) | **89.9±1.2** | **86.1±1.5** | **84.6±1.9** | **78.0±2.8** | **81.2±2.3** | **70.3±3.4** | 70.2±3.4 | **0.39±0.03** | **0.88±0.02** |
| Sequence (PLM)+Structure Token | 89.7±1.9 | 85.2±1.5 | 82.6±2.8 | 75.0±3.9 | 78.6±2.7 | 67.6±3.6 | 68.7±3.1 | 0.40±0.05 | 0.85±0.04 |
| Sequence (PLM)+Descriptors | 88.4±1.7 | 82.2±1.3 | 76.9±2.3 | 74.9±3.8 | 75.8±1.8 | 61.9±2.7 | 66.1±3.7 | 0.41±0.03 | 0.84±0.03 |
| VENUSVACCINE | 89.7±1.5 | 85.8±2.0 | 81.8±3.0 | 77.2±2.5 | 80.2±2.2 | 69.2±3.4 | **86.5±2.5** | 0.46±0.06 | 0.69±0.03 |

Table 13: Average performance of baseline methods in **Virus** immunogenicity prediction tasks from 10 random splits, with the standard deviation reported after ±.

| input module | AUC-ROC | Accuracy | Precision | Recall | F1 | MCC | TopK-30 | Cross Entropy | KS statistic |
|---|---|---|---|---|---|---|---|---|---|
| Sequence (One-hot) | 96.8±0.4 | 91.5±0.7 | 92.1±1.5 | 91.2±0.8 | 91.6±0.7 | 82.9±1.5 | 98.0±1.2 | 0.26±0.03 | 0.84 ±0.01 |
| | | | | Ankh | | | | | |
| Sequence (PLM) | 94.6±0.9 | 84.8±1.5 | 81.3±2.3 | 91.0±1.3 | 85.8±2.5 | 70.1±2.9 | 75.4±2.5 | 0.34±0.02 | 0.98±0.01 |
| Sequence (PLM)+Structure Token | **97.9±0.6** | **93.7±0.6** | **95.5±0.7** | 92.2±1.1 | **93.8±0.7** | **87.5±1.2** | 88.9±1.8 | **0.20±0.03** | **0.99±0.00** |
| Sequence (PLM)+Descriptors | 96.1±0.7 | 88.1±1.0 | 85.2±1.6 | 92.6±2.0 | 88.8±2.1 | 76.4±2.0 | 81.3±2.1 | 0.28±0.02 | 0.99±0.01 |
| VENUSVACCINE | 97.7±0.4 | 92.5±0.6 | 91.4±1.3 | **95.5±1.3** | 92.8±0.7 | 85.2±1.2 | **99.5±0.4** | 0.22±0.02 | 0.87±0.02 |
| | | | | ESM | | | | | |
| Sequence (PLM) | 96.0±0.8 | 90.0±1.1 | 89.0±1.7 | 91.6±2.1 | 90.3±2.1 | 80.0±1.9 | 81.5±2.0 | 0.26±0.03 | 0.99±0.01 |
| Sequence (PLM)+Structure Token | 97.5±0.6 | 93.7±0.9 | **95.8±0.8** | 91.7±1.0 | 93.7±0.9 | 87.4±1.7 | 89.1±1.8 | 0.29±0.05 | **1.00±0.01** |
| Sequence (PLM)+Descriptors | 96.9±0.5 | 91.2±0.8 | 91.5±1.2 | 90.9±1.7 | 91.2±1.7 | 82.3±1.8 | 84.6±1.3 | **0.23±0.02** | 0.99±0.00 |
| VENUSVACCINE | **97.9±0.5** | **94.5±0.6** | 95.7±0.7 | **93.6±1.0** | **94.7±0.6** | **89.2±1.2** | **98.9±0.5** | 0.29±0.03 | 0.90±0.01 |
| | | | | ProtBert | | | | | |
| Sequence (PLM) | 97.2±0.7 | 91.4±1.2 | 89.3±1.4 | 94.2±2.3 | 91.7±2.3 | 82.9±2.3 | 84.8±2.3 | 0.23±0.03 | 0.99±0.01 |
| Sequence (PLM)+Structure Token | 96.8±0.5 | 90.8±0.9 | 90.4±0.9 | 91.7±1.5 | 91.0±1.7 | 81.6±1.9 | 82.9±1.7 | 0.23±0.02 | 0.99±0.01 |
| Sequence (PLM)+Descriptors | 96.9±0.5 | 91.5±1.0 | 89.4±1.5 | **94.6±2.1** | 91.9±2.1 | 83.0±2.1 | 85.3±2.1 | 0.24±0.03 | **0.99±0.01** |
| VENUSVACCINE | **97.5±0.6** | 91.6±0.9 | **97.0±0.8** | 93.5±1.4 | **92.9±0.8** | 85.6±1.5 | **98.7±0.9** | 0.23±0.02 | 0.88±0.02 |

Table 14: Average performance of baseline methods in **Tumor** immunogenicity prediction tasks from 10 random splits, with the standard deviation reported after ±.

| input module | AUC-ROC | Accuracy | Precision | Recall | F1 | MCC | TopK-30 | Cross Entropy | KS statistic |
|---|---|---|---|---|---|---|---|---|---|
| Sequence (One-hot) | 73.1±4.8 | 65.5±4.6 | 56.4±6.3 | 64.5±6.5 | 60.0±5.2 | 30.3±9.4 | 66.5±6.1 | 0.90±0.13 | 0.39 ±0.07 |
| | | | | Ankh | | | | | |
| Sequence (PLM) | 65.9±3.3 | 60.9±2.0 | 66.9±1.0 | 75.3±1.0 | 71.3±1.2 | 50.2±3.3 | 33.4±3.3 | 0.67±0.00 | 0.51±0.06 |
| Sequence (PLM)+Structure Token | 83.1±3.0 | 75.9±2.9 | 66.9±4.5 | 80.5±5.5 | 72.9±6.1 | 52.3±6.1 | 60.0±6.2 | 0.51±0.05 | **0.81±0.07** |
| Sequence (PLM)+Descriptors | 67.0±6.3 | 62.7±3.2 | 65.7±0.0 | 80.2±0.0 | 73.2±1.2 | 53.4±2.3 | 33.5±0.0 | 0.66±0.01 | 0.49±0.09 |
| VENUSVACCINE | **85.2±1.9** | **77.7±2.1** | 70.2±6.1 | **83.0±3.8** | **74.5±3.3** | **55.8±4.4** | **61.6±2.6** | **0.48±0.03** | 0.77±0.03 |
| | | | | ESM | | | | | |
| Sequence (PLM) | 81.5±4.3 | 70.4±3.6 | 60.0±4.7 | 75.7±3.4 | 66.8±6.8 | 41.8±9.1 | 52.5±4.0 | **0.51±0.04** | 0.73±0.07 |
| Sequence (PLM)+Structure Token | **83.4±3.4** | 70.6±4.2 | 58.0±5.5 | **81.7±5.6** | **67.7±5.0** | 44.3±8.0 | **58.6±7.6** | 0.51±0.05 | **0.74±0.07** |
| Sequence (PLM)+Descriptors | 78.4±2.0 | 68.6±2.6 | 59.8±5.1 | 73.9±4.0 | 65.9±3.3 | 38.0±4.8 | 44.3±3.5 | 0.55±0.02 | 0.72±0.07 |
| VENUSVACCINE | 82.2±3.5 | **73.7±3.1** | **67.8±3.5** | 66.8±5.4 | 67.1±4.7 | **45.5±5.9** | 58.2±5.6 | 1.57±0.02 | 0.68±0.07 |
| | | | | ProtBert | | | | | |
| Sequence (PLM) | 69.7±3.0 | 59.9±3.8 | 51.8±6.5 | 64.7±5.4 | 57.3±5.2 | 20.9±7.8 | 36.5±4.7 | 0.61±0.02 | 0.61±0.07 |
| Sequence (PLM)+Structure Token | 80.4±3.0 | 70.5±3.3 | 59.6±6.0 | 76.4±6.1 | 66.8±4.0 | **45.9±6.0** | 55.9±5.2 | **0.53±0.05** | **0.67±0.06** |
| Sequence (PLM)+Descriptors | 75.7±3.3 | 69.5±3.7 | 58.9±6.5 | 76.6±5.9 | 66.5±5.2 | 40.4±7.5 | 46.7±7.8 | 0.61±0.04 | 0.66±0.07 |
| VENUSVACCINE | **81.0±3.2** | **71.3±3.5** | 59.6±5.4 | **78.1±3.9** | **67.1±5.3** | 43.9±8.1 | **67.8±7.3** | 0.58±0.07 | 0.52±0.08 |

Table 15: Model prediction performance on the first case study of Helicobacter pylori.

| Model | Predicted Positive | True Positive | Random Selection | Recall (%) | fold-enrichment |
|---|---|---|---|---|---|
| Random Forest | 211 | 11 | 1.25 | 100.00 | 8.81 |
| Gradient Boosting | 273 | 11 | 1.62 | 100.00 | 6.81 |
| XGBoost | 287 | 11 | 1.70 | 100.00 | 6.47 |
| SGD | 25 | 0 | 0.15 | 0.00 | 0.00 |
| Logistic Regression | 0 | 0 | 0.00 | 0.00 | 0.00 |
| MLP | 227 | 4 | 1.34 | 36.36 | 2.98 |
| KNN | 284 | 10 | 1.68 | 90.91 | 5.95 |
| VENUSVACCINE | 123 | 11 | 0.73 | 100.00 | 15.11 |

Table 16: Detailed likelihood predicted for the 11 determined immunogens under each model.

| Protein | Random Forest | Gradient Boosting | XGBoost | SGD | Log Reg | MLP | SVM | KNN | VENUSVACCINE |
|---|---|---|---|---|---|---|---|---|---|
| P69997 | 0.93 | 0.910173 | 0.998961 | 0.311594 | 0.412161 | 0.623967 | 0.689306 | 1 | 0.878434 |
| Q9ZN50 | 0.87 | 0.816795 | 0.997187 | 0.253575 | 0.415796 | 0.492811 | 0.698149 | 1 | 0.96182 |
| Q9ZKE6 | 0.74 | 0.586535 | 0.988803 | 0.057802 | 0.394834 | 0.072825 | 0.820259 | 1 | 0.82105 |
| Q9ZKW5 | 0.84 | 0.950726 | 0.999409 | 0.429902 | 0.417487 | 0.844049 | 0.72135 | 1 | 0.993692 |
| Q9ZKX5 | 0.78 | 0.600814 | 0.986793 | 0.154398 | 0.396191 | 0.41584 | 0.712237 | 0.5 | 0.738437 |
| Q9ZMJ1 | 0.72 | 0.564479 | 0.985005 | 0.099751 | 0.396345 | 0.341039 | 0.708772 | 1 | 0.725007 |
| Q9ZL47 | 0.83 | 0.765695 | 0.994611 | 0.135861 | 0.402306 | 0.233893 | 0.645672 | 1 | 0.82569 |
| Q9ZLT1 | 0.66 | 0.673526 | 0.860356 | 0.371664 | 0.412658 | 0.811522 | 0.771533 | 1 | 0.986235 |
| Q9ZN37 | 0.84 | 0.749941 | 0.99151 | 0.153825 | 0.398239 | 0.338815 | 0.792457 | 1 | 0.659044 |
| Q9ZLJ8 | 0.61 | 0.911371 | 0.989448 | 0.339596 | 0.410059 | 0.715878 | 0.574296 | 1 | 0.978299 |
| P0A0R4 | 0.94 | 0.880224 | 0.991948 | 0.168698 | 0.401071 | 0.454899 | 0.335555 | 1 | 0.881635 |

Table 17: Prediction Details of SARS-COV-2 Targets by VENUSVACCINE.

| NCBI ID | Protein Name | predicted label | predicted probability |
|---------|--------------|:---------------:|:---------------------:|
| YP_009724390.1 | surface glycoprotein | 1 | 1.000 |
| YP_009742617.1 | nsp10 | 1 | 1.000 |
| YP_009742616.1 | nsp9 | 1 | 1.000 |
| YP_009742609.1 | nsp2 | 1 | 0.999 |
| YP_009742612.1 | 3C-like proteinase | 1 | 0.998 |
| YP_009725312.1 | nsp11 | 1 | 0.997 |
| YP_009724396.1 | ORF8 protein | 1 | 0.997 |
| YP_009742610.1 | nsp3 | 1 | 0.997 |
| YP_009724397.2 | nucleocapsid phosphoprotein | 1 | 0.995 |
| YP_009742614.1 | nsp7 | 1 | 0.994 |
| YP_009724395.1 | ORF7a protein | 1 | 0.941 |
| YP_009725307.1 | RNA-dependent RNA polymerase | 1 | 0.864 |
| YP_009725310.1 | endoRNAse | 1 | 0.768 |
| YP_009742611.1 | nsp4 | 0 | 0.494 |
| YP_009724391.1 | ORF3a protein | 0 | 0.463 |
| YP_009742608.1 | leader protein | 0 | 0.150 |
| YP_009725308.1 | helicase | 0 | 0.140 |
| YP_009742613.1 | nsp6 | 0 | 0.107 |
| YP_009742615.1 | nsp8 | 0 | 0.047 |
| YP_009725309.1 | 3'-to-5' exonuclease | 0 | 0.022 |
| YP_009725311.1 | 2'-O-ribose methyltransferase | 0 | 0.000 |
| YP_009724393.1 | membrane glycoprotein | 0 | 0.000 |
| YP_009724392.1 | envelope protein | 0 | 0.000 |
| YP_009724394.1 | ORF6 protein | 0 | 0.000 |
| YP_009725318.1 | ORF7b | 0 | 0.000 |
| YP_009725255.1 | ORF10 protein | 0 | 0.000 |

Figure 6: Convergence speed of VENUSVACCINE versus the vanilla PLMs on **ImmunoDB-Virus**.