# OpenReview forum: "Immunogenicity Prediction with Dual Attention Enables Vaccine Target Selection"
_ICLR.cc/2025/Conference — ICLR 2025 Poster_

### Official Review · Reviewer_BQLB · 2024-11-01

**Soundness:** 2
**Presentation:** 2
**Contribution:** 1
**Rating:** 6
**Confidence:** 5

**Summary:**

In the paper, the authors proposed a protein-language-model-based method called PROVACCINE for predicting the antigen from the amino acid sequence. They constructed the largest and most comprehensive antigen dataset through collecting published sequences. They further conducted cross-validation and some post-hoc analysis to show the effectiveness of the proposed model.

**Strengths:**

The paper conducted generalization experiments, which is good.
The comprehensive datasets, especially the positive sequence may work as the foundation for other work.

**Weaknesses:**

1. What I am mostly concerned with is the dataset construction. It seems that the authors construct the positive dataset from the literature, which is great, but the negative data are constructed with some other computational tools. This will lead to a serious problem: if the golden standard is created using BLAST+VAXIJEN, then if we use the combination of the two will be enough, why do we bother to train a model on top of it?

2. Continuing from 1, the authors should add the baseline method using their way to construct the training dataset, they will get a perfect score for all metrics. The logic for building negative samples is very weird and then making the model unreliable.

3. In section 5.2.1, the authors should also consider splitting the data based on the sequence similarity. If the training and testing data share very high sequence similarity, it's very likely that the model just memorizes the training and can not generalize.

4. I see XGboost and KNN are good from Figure 4 -- suggesting the novel antigens share high similarity with known ones. Actually XGBoost seems to be perfect and much better than Provaccine. The proposed model did not show superiority than the traditional similarity-based method. Moreover, as majority of these 1, 858 sequences lack immunogenicity ground truth labels, how can we ensure the enrichment is a good metric?

5. In section 5.3.2, it's unclear whether the spike protein already exist in the training dataset

6. Continuing from 6, section 5.3.2 is too rough -- it's very easy to identify the spike protein as well as the RBD as the antigen, the results can not support the effectiveness of the proposed model.

Taken together, the proposed method did not show enough novelty, and the performance is not as good as the similarity-based method.

**Questions:**

See weakness above

---

> ### Author Response · Authors · 2024-11-22
> **Response (1)**
>
> Thank you for the review and recognize the novelty of our work. Below we respond to your concerns and questions point-by-point.
>
> **Weaknesses**:
>
>
> 1. **Dataset Construction**: We agree with your concerns regarding the construction of negative samples. This method indeed has notable flaws, and assuming that a combination of BLAST and VaxiJen serves as a gold standard is certainly not ideal. However, the lack of negative instances is a well-known issue in many scientific problems, and we chose to adhere to an acceptable, though suboptimal, method that has been used in prior publications (Dimitrov et al., 2020; Sotirov & Dimitrov, 2024; Doneva & Dimitrov, 2024). We have addressed this discussion in the manuscript (see lines 176-179 on page 4). In the revision, **following Reviewer G2xM’s suggestion, we explored alternative data construction methods** by augmenting negative instances from IEDB database validated more than two negative B cell assays, which make the negative data set more convincing. We evaluated the results of this new construction on both the baselines and our method, as shown below:
>
> |        Model        | |       |    Bacteria   |       |      | |       |     Virus  |       |      | |       |  Tumor     |       |      |
> |:-------------------:|:--------:|:-----:|:-----:|:-----:|:----:|:-----:|:-----:|:-----:|:-----:|:----:|:-----:|:-----:|:-----:|:-----:|:----:|
> |                     |    ACC   |  T30  |  MCC  |   F1  |  KS  |  ACC  |  T30  |  MCC  |   F1  |  KS  |  ACC  |  T30  |  MCC  |   F1  |  KS  |
> |    Random Forest    |   81.1   | 77.7  | 57.1  | 69.7  | 0.60 | 88.3  | 98.7  | 76.7  | 88.5  | 0.80 | 66.8  | 63.5  | 27.0  | 46.4  | 0.42 |
> |  Gradient Boosting  |   80.7   | 75.9  | 56.8  | 71.2  | 0.62 | 86.0  | 97.3  | 72.3  | 86.5  | 0.74 | 65.4  | 60.0  | 26.0  | 53.1  | 0.38 |
> |       XGBoost       |   80.8   | 76.1  | 57.0  | 71.1  | 0.61 | 89.1  | 98.8  | 78.2  | 89.2  | 0.80 | 69.2  | 62.2  | 33.7  | 56.1  | 0.40 |
> |         SGD         |   78.6   | 72.4  | 51.0  | 65.3  | 0.56 | 77.3  | 88.0  | 56.0  | 74.0  | 0.66 | 52.6  | 50.9  | 29.6  | 62.2  | 0.31 |
> | Logistic Regression |   65.2   | 67.3  |  1.8  |  0.5  | 0.47 | 61.4  | 82.9  | 35.6  | 71.9  | 0.61 | 60.4  | 45.2  |  0.0  |  0.0  | 0.25 |
> |         MLP         |   78.1   | 71.6  | 51.5  | 68.1  | 0.57 | 85.0  | 90.6  | 70.5  | 85.7  | 0.71 | 60.4  | 39.1  |  0.0  |  0.0  | 0.26 |
> |         SVM         |   56.7   | 57.3  | 18.9  | 53.0  | 0.33 | 61.6  | 88.7  | 25.1  | 67.3  | 0.53 | 51.4  | 61.3  | 15.2  | 57.3  | 0.33 |
> |         KNN         |   80.4   | 73.8  | 55.3  | 68.6  | 0.52 | 87.4  | 86.8  | 74.8  | 87.3  | 0.75 | 57.4  | 49.6  | 10.8  | 45.3  | 0.12 |
> |         Vexi-DL       |   68.1   | 55.5  | 41.7  | 66.9  | 0.42 | 65.3  | 62.2  | 33.4  | 72.9  | 0.29 | 54.9  | 41.3  |  8.5  | 47.7  | 0.10 |
> |      VaxiJen2.0     |   75.7   | 62.2  | 54.6  | 72.1  | 0.57 | 82.0  | 74.8  | 66.6  | 84.2  | 0.64 |   -   |   -   |   -   |   -   |   -  |
> |      VaxiJen3.0     |   83.3   | 58.7  | 63.2  | 75.1  | 0.63 | 68.1  | 62.5  | 42.3  | 75.4  | 0.35 | 39.0  | 35.7  |  0.0  | 55.9  | 0.00 |
> |      VirusImmu      |     -    |   -   |   -   |   -   |   -  | 58.8  | 59.5  | 18.1  | 63.6  | 0.17 |   -   |   -   |   -   |   -   |   -  |
> |   ProVaccine-Ankh   |   82.3   | 78.5  | 60.3  | 73.3  | 0.64 | 92.2  | 99.7  | 84.3  | 92.3  | 0.85 | 76.9  | 73.5  | 55.0  | 73.5  | 0.61 |
> |   ProVaccine-ESM2   |   80.6   | 77.0  | 57.0  | 71.7  | 0.66 | 90.3  | 99.5  | 80.6  | 90.6  | 0.82 | 74.0  | 61.7  | 46.1  | 68.2  | 0.58 |
> | ProVaccine-ProtBert |   84.5   | 84.5  | 65.9  | 77.0  | 0.66 | 91.4  | 97.4  | 82.8  | 91.2  | 0.84 | 71.5  | 68.7  | 44.7  | 67.6  | 0.54 |
>
>
>
> 2. **Negative Label Assignment Using Different Baseline Methods**: As explained above, our use of VaxiJen to define negative samples follows existing publications. Introducing an advantageous bias for a baseline method (*i.e.*, VaxiJen) holds no benefit for our work, and empirically VaxiJen did not achieve top performance under this construction method. The new data construction approach we tried during the revision included more negative samples directly collected from IEDB and publications, thus reducing bias toward any model. As shown in the previous results, our method still achieves top performance, demonstrating a clear advantage of ProVaccine over existing methods.

---

> ### Author Response · Authors · 2024-11-22
> **Response (2)**
>
> 3. **Data Splitting**: In the initial submission, we only checked that there are no 100% same sequences exist in the dataset. Following the suggestion, we applied an additional redundancy removal step between all training and test samples with a 90% similarity cut-off to avoid potential data leakage. We test the performance on Immuno-Bateria and use ESMFold2 for structure prediction. After this modification, we observed a reasonable decrease in performance across all models, including the baseline models. Despite this challenge, ProVaccine continues to outperform other models and demonstrates its robustness and effectiveness.
>
>  | Model | Accuracy |  TopK  |   MCC  |   F1   | KS-statistic |
> |:-:|:-:|:-:|:-:|:-:|:-:|
> | Random Forest | 89.6     | 100.0 | 79.2  | 90.2  | 0.847        |
> | Gradient Boosting | 87.8     | 100.0 | 75.5  | 88.6  | 0.802        |
> | XGBoost | 92.7     | 100.0 | 85.4  | 93.2  | 0.890        |
> | SGD                                        | 82.0     | 92.1  | 65.4  | 81.3  | 0.714        |
> | Logistic Regression                        | 62.1     | 90.5  | 34.0  | 73.5  | 0.661        |
> | MLP                                        | 85.4     | 91.2  | 70.6  | 86.1  | 0.738        |
> | SVM                                        | 78.4     | 92.9  | 57.8  | 77.9  | 0.614        |
> | KNN                                        | 90.1     | 93.1  | 80.3  | 90.5  | 0.803        |
> | Vexi-DL                                       | 65.4     | 62.2  | 33.4  | 72.9  | 0.285        |
> | VaxiJen2.0                                 | 82.0     | 74.8  | 66.6  | 84.2  | 0.641        |
> | VaxiJen3.0                                 | 68.2     | 62.5  | 42.4  | 75.4  | 0.349        |
> | VirusImmu                                  | 58.8     | 59.6  | 18.1  | 63.6  | 0.175        |
> | ProVaccine-Ankh | 91.6     | 100.0 | 84.0  | 91.7  | 0.906        |
> | ProVaccine-ESM2 | 92.4 | 100.0 | 85.0  | 92.8  | 0.876        |
> | ProVaccine-ProtBert    | 94.9     | 100.0 | 89.8  | 95.0  | 0.920        |
>
> 4. **Comparison between XGBoost and ProVaccine**:  (1) Would you mind provide additional explanations regarding “*the novel antigens share high similarity with known ones*”? We are unsure about the evidence that supports this statement. (2) We respectively disagree with your statement that “*Actually XGBoost seems to be perfect and much better than Provaccine*” for two reasons, following the observations we concluded in Section 5.3.1, First, the majority of antigens in nature are non-immunogenic, meaning they should be assigned a negative label in our prediction task. Therefore, **an ideal predictive model should assign probabilities closer to 0 for most data points**, rather than distributing them more uniformly across the 0-1 range (as observed in many other baseline methods) or concentrating them in a very narrow region (as seen in logistic regression). The KDE plot in Figure 4 shows clearly that only ProVaccine manages to approximate this behavior. Secondly, the time and financial costs of biological experiments are significant. Therefore, in addition to capturing true positives, **it is preferable for a model to minimize false positive predictions**. As shown in Table 10 of the Appendix, our method assigns the fewest positive labels (i.e., the fewest PVCs) while successfully identifying all 11 true positives. This demonstrates that our method can identify positive immunogens at a lower experimental cost.
>
> 5. **The Existence of Spike Proteins in the Training Dataset?**: We examined the dataset and confirmed that spike proteins (as well as all the proteins assessed in the two case studies) are not present in the training dataset. In the revision, we add explicit statement regarding the exclusion of these proteins to avoid confusion.
>
> 6. **Difficulty of Identifying the Spike proteins and RBD regions and model effectiveness**: We would like to clarify two important points in response to your concerns: (1) While the spike protein was successfully predicted as the top candidate, **the model had no prior exposure to the SARS-CoV-2 spike protein** (as this protein was never seen during training). This highlights the model’s ability to generalize knowledge to new spike proteins. Although the spike protein is a prominent candidate in current vaccines, the model's ability to make this prediction without explicit training on SARS-CoV-2 data underscores its reliability. (2) We designed the analysis of **identifying potential RBD regions to demonstrate the explainability and potential usefulness of the attention scores produced by the model**. Instead, we assess the model’s effectiveness through previous experiments, such as overall performance comparisons on benchmark datasets.
>
> We believe the above responses clearly address your concerns and clarify the misunderstandings in the previous submission. We look forward to your feedback and are happy to continue the discussion if there are any remaining issues.

---

> > ### Comment · Reviewer_BQLB · 2024-11-25
> >
> > Thank you for the detailed experiments! These experiments solved my most concerns. Before I raise my score, I wonder if it's possible to set the similarity cut-off at 0.4. Sequence similarity higher than 0.4 are generally considered as homologous proteins. I would like to check the model performance under such extreme conditions to get hints about the OOD ability.

---

> > > ### Author Response · Authors · 2024-11-26
> > >
> > > Thank you for your insightful comments and for recognizing the value of our supplementary experiments and efforts to address your concerns. Following your suggestions, we have added the evaluation on the dataset under a 40% sequence identity threshold, as presented in the table below:
> > >
> > > | Model                |    |    | **Bacteria**    |  |   |   |    | **Virus**    |  |    |   |    | **Tumor**    |  |    |
> > > |:---:|:---:|:---:|:---:|:---:|:---:|:---:|:---:|:---:|:---:|:---:|:---:|:---:|:---:|:---:|:---:|
> > > |                          | ACC                | T30    | MCC    | F1     | KS     | ACC | T30    | MCC    | F1     | KS     | ACC                | T30    | MCC    | F1     | KS     |
> > > | Random Forest            | 85.5              | 84.1 | 68.2  | 78.8   | 0.71    | 89.6 | 94.3   | 79.4   | 88.6   | 0.81    | 67.4              | 67.6   | 31.3   | 50.2   | 0.42    |
> > > | Gradient Boosting        | 86.4          | 82.7   | 70.3 | 80.8 | 0.74 | 78.3 | 91.4   | 55.6   | 77.0   | 0.74    | 66.3              | 65.2   | 29.3   | 55.6   | 0.41    |
> > > | XGBoost                  | 82.0              | 82.7   | 61.4   | 75.5   | 0.66    | 90.8 | 97.1 | 80.7  | 89.4   | 0.86    | 70.7              | 69.0   | 38.8   | 59.5   | 0.43    |
> > > | SGD                      | 72.2              | 74.6   | 38.0   | 41.7   | 0.59    | 80.0 | 92.9   | 61.4   | 75.4   | 0.79    | 55.1              | 48.6   | 30.9   | 64.7   | 0.32    |
> > > | Logistic Regression      | 63.5              | 63.6   | 0.0    | 0.0    | 0.45    | 59.2 | 82.9   | 31.2   | 68.9   | 0.73    | 58.0              | 48.1   | 0.0    | 0.0    | 0.24    |
> > > | MLP                      | 78.5              | 75.0   | 53.5   | 70.3   | 0.62    | 88.8 | 92.9   | 76.8   | 86.9   | 0.83    | 58.0              | 45.2   | 0.0    | 0.0    | 0.27    |
> > > | SVM                      | 54.9              | 61.4   | 20.7   | 56.2   | 0.38    | 77.9 | 84.3   | 53.7   | 75.4   | 0.68    | 55.4              | 70.5 | 20.5  | 61.0   | 0.40    |
> > > | KNN                      | 84.3              | 79.1   | 65.6   | 77.2   | 0.64    | 91.7 | 97.1 | 83.4 | 90.6 | 0.84    | 59.6              | 50.5   | 16.1   | 49.6   | 0.18    |
> > > | ProVaccine-Ankh          | 87.6 | 89.1 | 73.1 | 82.2 | 0.75 | 87.1 | 95.7   | 74.7   | 84.4   | 0.94 | 81.4          | 84.3 | 63.3 | 79.3 | 0.67 |
> > > | ProVaccine-ESM2          | 82.3 | 83.2   | 62.9   | 77.2   | 0.73    | 92.1 | 100.0 | 84.5 | 91.2 | 0.94 | 73.9          | 62.4   | 45.7   | 67.5   | 0.56 |
> > > | ProVaccine-ProtBert      | 86.9          | 88.6 | 71.7 | 81.8 | 0.82 | 92.9 | 100.0 | 85.9 | 92.0 | 0.96 | 73.4              | 73.3 | 47.8 | 70.9 | 0.56 |
> > >
> > > Upon implementing the 40% sequence identity cut-off, we observe that both our model and the baseline exhibit performance improvements across most metrics. We attribute this improvement to the removal of proteins with high sequence similarity but distinct immunogenicity labels. Additionally, this redundancy removal step may help eliminate mislabeled negative samples. For instance, a protein with a low sequence identity to a positive protein is more likely to be non-immunogenic than one with a high sequence identity to the positive protein. Based on this, the fact that our model continues to demonstrate top performance on the updated test set further highlights its predictive capabilities. These results have been incorporated into Table 7 of the revised Appendix.

---

> > > ### Author Response · Authors · 2024-11-29
> > >
> > > Thank you for your updated rating and recognition of our efforts. We are grateful for your insightful comments and engagement in improving our manuscript!

---

### Official Review · Reviewer_dRH5 · 2024-11-03

**Soundness:** 2
**Presentation:** 3
**Contribution:** 3
**Rating:** 6
**Confidence:** 4

**Summary:**

This paper introduced PROVACCINE, a novel deep-learning solution for immunogenicity prediction in reverse vaccinology, designed to identify candidate vaccines that trigger protective immune responses. The authors also constructed a comprehensive immunogenicity dataset with over 9,500 antigen sequences from various pathogens. PROVACCINE employs a dual attention mechanism, integrating pre-trained representations of protein sequences and structures, and outperforms existing methods. The paper provided a comprehensive overview of the problem, related work, and the proposed methodology.

**Strengths:**

1. This paper proposed a deep learning supervised learning framework with a dual attention mechanism to address the key challenges of immunogenicity prediction in vaccine development.
2. This paper provided a valuable benchmark for future research in vaccine development and immunogenicity prediction.

**Weaknesses:**

1. The work is innovative to construct a new benchmark dataset, but they didn't show novelty in algorithm development. The authors need to prove their methods on existed dataset, in addition to their own dataset.
2. The dual-attention mechanism, while sophisticated, may increase the computational cost and complexity of the model significantly. The paper does not thoroughly compare this method against simpler architectures to show that the added complexity justifies the performance gains.
3. The model’s performance is similar to XGBoost on the largest dataset, Immuno-Virus (1952/1762 positive/negative instances). However, the main improvement over XGBoost is on the smallest tumor dataset (300/477 positive/negative instances). Usually, DL is strong for big dataset and weak for small dataset, in contrary to this study. This discrepancy raised questions about the reliability and robustness of the model's performance.
3. A major contribution of this work is the construction of a data set for immunogenicity prediction and vaccine development. It’s insufficient to discuss data quality, noise, or variability, particularly for immunogenic labels sourced from different repositories. Moreover, authors should demonstrate that the new dataset is superior to existing datasets.
4. The authors should include ablation comparisons of sequence and structural features?
5. This paper only compares web server methods, lacking comparisons with other deep learning frameworks.

**Questions:**

The authors need to prove their methods on existing datasets. They also need to dig out why the model shows the largest improvement on the smallest dataset. The case study also needs to discuss about reason, instead of the performance only.

---

> ### Author Response · Authors · 2024-11-22
> **Response (1)**
>
> Thank you for acknowledging the novelty and contributions of our work in terms of the framework and benchmark dataset. We notice that we need to first emphasize our contributions and clarify that proposing a new benchmark for  immunogenicity prediction is not the only major contribution of our study. To be precise:
> 1. **Novel Algorithmic Approach**: Our algorithm represents a significant departure from traditional machine learning in immunogenicity prediction. We introduce a pioneering multimodal deep learning-based model that integrates sequence, structural, and physicochemical data. The proposed model achieves top performance on benchmark datasets and show promising generalizabiltiy and interpretability in case study analyses.
> 2. **Extensive Dataset Curation**: We present an extensive, curated dataset of antigens from various sources. To the best of our knowledge, it offers the largest and most comprehensive training and testing benchmarks for the immunogenicity prediction community and serving.
> 3. **Comprehensive Assessment Methodology**: We designed the evaluation protocols based on meaningful criteria that are crucial to biologists and significantly relevant to immunological research. In addition to commonly used metrics in computer science (such as accuracy and recall) to assess the model's performance on the test set, we specifically included two case studies to evaluate the practical value of the model for vaccine development. This analytical approach also provides a novel validation framework for future research that may lack experimental conditions.
>
> These concludes were also described in manuscript (line 93-98, in page2). For the other weaknesses and questions you raised, below is our point-by-point response.
>
> **Weaknesses**:
>
>
> 1. **Testing on Existing Benchmark Datasets**: We understand that testing on benchmarks used in previous studies is usually an essential validation step when proposing a new algorithm. However, we did not include additional comparison on other datasets for two main reasons:
> (1) **All benchmark datasets used in existing studies correspond to a subset of our dataset**, to the best of our knowledge. This is different from other well-defined and largely explored tasks in machine learning (such as developing graph convolutional algorithms and testing on citation networks), As introduced in Section 3 of the manuscript, we incorporated data from a broader range of literature, included more diverse species, and extracted and processed updated records from public databases.
> (2) **All our data is sourced from published studies and adheres to data processing principles from previous research** (such as Dimitrov et al., 2020; Sotirov & Dimitrov, 2024; and Doneva & Dimitrov, 2024). A similar example of study that only tested on their new benchmark is ProteinGym (Notin et al., 2024), which consolidates multiple existing benchmarks plus additional benchmarks from different literature. As a result, their testings were also focused on their proposed larger, and more comprehensive benchmark dataset. We thus believe it is sufficient to evaluate baseline methods on our proposed new benchmark dataset.
>
>
> 2. **Computational Cost of the Algorithm**: To address this problem, we added an analysis of computational cost comparison in Figure 6 (in the Appendix) in the revision. We compared the training curves of ProVaccine and three vanilla PLMs on the VirusBinary dataset. It is evident that **ProVaccine exhibits a much faster convergence rate, approximately 2-3 times faster than the corresponding vanilla PLM methods**. Meanwhile, the per-epoch runtime of ProVaccine's is comparable to that of vanilla PLMs. For instance, training ESM2-650M for one epoch takes 2 hours and 52 minutes, while ProVaccine-ESM2-650M requires 2 hours and 56 minutes per epoch.

---

> ### Author Response · Authors · 2024-11-22
> **Response (2)**
>
> 3. **Improvement over XGBoost on Virus vs Tumor**: (1) Regarding the statement, “*The model’s performance is similar to XGBoost on the largest dataset, Immuno-Virus*”, we assume this observation is based on Table 1 of the initial submission. We respectfully disagree with this statement because, except for T30 (where our DL methods ranked second), our ProVaccine consistently ranked in the top three for all other metrics. Furthermore, on the updated dataset (with improvements in negative sample construction and the use of AF2-based predicted structures, as suggested by reviewer G2xM), the performance gap between our method and XGBoost has further widened.
> (2) It is commonly observed that DL methods perform better on larger datasets due to their capacity to learn more parameters, which typically requires a larger volume of data for supervised learning tasks. However, in our case, we first utilize pre-trained PLM modules and a structure tokenization module to learn effective representations of protein sequences and structures. We greatly appreciate your observation of this phenomenon, which may indicate an additional novelty of our approach, i.e., its capability to perform robustly even with relatively smaller sample sizes. Unfortunately, the limited time and computational resources during the rebuttal phase restrict our ability to validate this finding. Nevertheless, this could be a promising direction for future work.
>
> 4. **Quality and Superiority of the New Benchmark Dataset**: We added a paragraph at the end of Section 3 to explicitly summarize the superiority of our new benchmark dataset compared to existing ones. In short, we provide larger and more comprehensive cross-species datasets with more reliable data processing and labeling procedures. For the various sources used to collect raw data, we conducted manual checks by immunology experts to minimize the risk of mislabelling issues. However, it is important to note that we cannot completely eliminate any potential noise originating from original experimental errors and variations in experimental technologies and environments. Such errors are inherent in other datasets derived from experimental sciences, and addressing them falls beyond the scope of our work.
>
> 5. **Ablation Study with Sequence and Structure Input**: The required ablation studies on different modules have been provided in Tables 7–9 in the appendix (uploaded as supplementary material). These studies include experiments with the following inputs: PLM-encoded sequence only, PLM-encoded sequence + atomic tokenization, PLM-encoded sequence + physiochemical descriptors, and the complete version combining PLM-encoded sequence (finer-level information) + atomic tokenization (coarser-level information) + physiochemical descriptors. Furthermore, following your suggestion, we conducted an additional experiment that directly predicted the information from one-hot encoded sequences. The results have been updated in the same tables in the revised appendix.
>
>
> 6. **Comparison with Other DL Methods**: To the best of our knowledge, there are no existing deep learning-based solutions for immunogenicity prediction, which is a research gap we have identified and mentioned in the manuscript. In our experiments, we included similar comparisons wherever possible. For instance, ProVaccine incorporates different base models (Ankh, ESM2, and ProtBert). Additionally, our ablation study, where we consider inputs only the PLM embedding, can be roughly regarded as a comparison with existing PLM-based models.
>
>
> **Questions**:
>
> 1. **Model Performance on Existing Datasets**: See response to weakness 1.
>
> 2. **Prediction Results on Virus and Tumor**: See response to weakness 3.
>
> 3. **Additional Analysis on Case Studies**: Thank you for the suggestion. In the revised manuscript, we have included an analysis of the factors contributing to the model's predictions, such as the characteristics of the antigens, the role of multimodal representations, and the impact of the dual-attention mechanism.
>
> In conclusion, we believe our work makes significant contributions not only in constructing a new benchmark dataset but also in advancing the algorithmic frontier of immunogenicity prediction and establishing robust evaluation protocols. The efficiency of algorithm as well as the quality and superiority of the new benchmark were identified.
>
> We hope this response addresses your concerns and highlights the innovative and comprehensive nature of our research. Thank you again for the review and suggestions. We look forward to your feedback and are happy to continue the discussion if there are any remaining issues.

---

### Official Review · Reviewer_G2xM · 2024-11-04

**Soundness:** 3
**Presentation:** 3
**Contribution:** 4
**Rating:** 8
**Confidence:** 4

**Summary:**

The paper presents PROVACCINE, a deep learning model designed to predict the immunogenicity of proteins, which is a crucial task in vaccine development. The authors introduce a dual-attention mechanism within the model, enabling it to process multimodal representations of protein sequences, structures, and physicochemical properties. They also curate Immuno, an extensive dataset of over 9,500 antigen sequences and structures from various sources, to facilitate robust training and validation. The experimental results show that PROVACCINE outperforms traditional methods across multiple evaluation metrics, with specific validation studies on Helicobacter pylori and SARS-CoV-2. The post-hoc analyses further confirm PROVACCINE’s efficacy in identifying potential vaccine targets.

**Strengths:**

- In my knowledge, the ProVaccine tool is the first method to utilize all three modalities of sequence, structure, and amino-acid descriptors to predict immunogenicity with deep-learning.
- The authors have tried multiple PLMs to encode the sequences (ESM2, ANKH, and ProtBert) and structural embedding methods (ESM3, FoldSeek) to encode the structures.
- They have compiled a new dataset called IMMUNO (stratified by antigen type- viral, bacterial, and tumour), which should serve as a valuable resource for training and benchmarking new methods for researchers in the field.
- They have carried out a very comprehensive benchmarking and compared their method to appropriate baselines and also previous methods in literature. They also carried-out a cross-testing benchmarking of viral, bacterial, and tumour-specific models across datasets to assess the generalizability of the method.
- The post-hoc analyses on specific pathogens (Helicobacterpylori and SARS-CoV-2) provide additional evidence of the model's usability in applied research. The method clearly improves upon immunogenicity prediction accuracy (against previous SOTA) and thus provides a reliable framework for future research in computational vaccinology.

**Weaknesses:**

-  Availability of negative dataset is a known problem in this field. The authors have used VAXIJEN to classify sequences as non-antigens and then subsequently filtered the sequences based on sequence-homology to compile the negative dataset. This will bias the negative dataset for certain features which might coincide with the features used in other tools (not just VAXIJEN, AA descriptors used for immunogenicity predictions are generally common across various methods). This might overinflate the recall for the negative class. An alternative approach could be to randomize the order of the amino-acids in the sequence while making sure that the sequence-identity of the randomized sequence isn't similar to any sequence in the positive class. The authors can also look at the IEDB database for compiling the negative dataset as it has sequences of non-antigens from experimental assays (one can chose sequences which were negative in atleast two independent studies)

- For the structural embeddings, the authors have relied on ESMFold for the structure of the antigenic proteins. ESMFold, while much faster than AlphaFold, lags behind in accuracy (in the backbone atoms as well) to AlphaFold/RosettaFold. There isn't any justification provided for choosing these models. There needs to be more analysis done to show how does the structural model quality affect the performance of ProVaccine. The comparison can be done against experimental structures (from PDB), AlphaFold2/3 structures, and other structure prediction methods.

**Questions:**

- Was there any homology cut-off applied between the training and testing datasets for each class of the antigens? High sequence similarity b/w training and testing might over-inflate performance.

- Most of  the figures have a generic caption. The captions should explain the figures clearly and the results being shown in the figure (especially the figures showing the results and comparison). The results from the figures should also be discussed in the main text in detail.

- In the post-hoc analysis, it it possible to check how well will the model perform for non-immunogenic antigens in Helicobacter pylori (Fig 4). The sequences of the 11- experimentally verified immunogens can be randomized and then it can be checked where does the prediction for the randomized-sequence lie in terms of probability. The authors also need to describe how 'the probability of identifying known protective antigens through random guessing' was calculated? It's not clear from the current description.

-  Model interpretability plays an important role for any downstream analysis and experimental work (especially in screening vaccine candidates or selecting antigens for vaccine development). Can the weights of the attention modules be used for identifying the immunogenic region in the antigen? This could possibly be leveraged to identify epitopes that can elicit protective antibodies when developing vaccines. This could be an useful addition to the manuscript.

- How would the model handle variations in immunogenicity features across less common or novel pathogens that are not represented in the current dataset?

---

> ### Author Response · Authors · 2024-11-22
> **Response (1)**
>
> Thank you for acknowledging the contributions of our work and for providing detailed and constructive feedbacks to further strengthen it. In response to the weaknesses and questions you raised, we have updated the paper accordingly and provide a point-by-point reply below.

---

> ### Author Response · Authors · 2024-11-22
> **Response (2)**
>
> **Weaknesses**:
>
> 1. **Dataset Construction**:
> We agree with you regarding the inherent limitation in constructing negative instances in the dataset, although this method has been used in previous studies (Dimitrov et al., 2020; Sotirov & Dimitrov, 2024; Doneva & Dimitrov, 2024). In the revision, we explored the two alternative approaches to constructing negative samples that you suggested.
> * We first attempted to generate random amino acid (AA) sequences as negative samples, ensuring that their sequence identity with any positive samples from the same dataset was less than 30%. However, we identified two significant issues with this implementation. First, the predicted structures generally exhibited low folding confidence, with average pLDDT scores of lower than 50%. Second, when we applied these new negative samples to the training dataset while using the original test dataset (which contains negative samples derived from natural sequences), we observed that across all three datasets, the models achieved 100% accuracy during training but trivially predicted positive labels for all test samples. Our analysis suggests that when negative samples are generated randomly, the model primarily **learns to differentiate between "protein" and "non-protein" rather than distinguishing "immunogenic proteins" from "non-immunogenic proteins"**.
> * We also attempted to prepare negative samples from the IEDB database, identifying bacterial and viral non-antigens confirmed through experimental B-cell assays in Homo sapiens. We excluded tumor samples because the tumor-related records in IEDB do not indicate whether they are from human sources.
> Notably, all positive antigens in IEDB were already included in our existing collection. After preprocessing steps such as redundancy removal and filtering out low-quality data, 149 novel bacterial and 124 viral non-antigens remained.
> We incorporated these newly labeled negative data into the existing dataset and updated the performance on both our model and the baseline models. The updated results are provided in Table 1 as follows:
>
> |        Model        | |       |    Bacteria   |       |      | |       |     Virus  |       |      | |       |  Tumor     |       |      |
> |:-------------------:|:--------:|:-----:|:-----:|:-----:|:----:|:-----:|:-----:|:-----:|:-----:|:----:|:-----:|:-----:|:-----:|:-----:|:----:|
> |                     |    ACC   |  T30  |  MCC  |   F1  |  KS  |  ACC  |  T30  |  MCC  |   F1  |  KS  |  ACC  |  T30  |  MCC  |   F1  |  KS  |
> |    Random Forest    |   81.1   | 77.7  | 57.1  | 69.7  | 0.60 | 88.3  | 98.7  | 76.7  | 88.5  | 0.80 | 66.8  | 63.5  | 27.0  | 46.4  | 0.42 |
> |  Gradient Boosting  |   80.7   | 75.9  | 56.8  | 71.2  | 0.62 | 86.0  | 97.3  | 72.3  | 86.5  | 0.74 | 65.4  | 60.0  | 26.0  | 53.1  | 0.38 |
> |       XGBoost       |   80.8   | 76.1  | 57.0  | 71.1  | 0.61 | 89.1  | 98.8  | 78.2  | 89.2  | 0.80 | 69.2  | 62.2  | 33.7  | 56.1  | 0.40 |
> |         SGD         |   78.6   | 72.4  | 51.0  | 65.3  | 0.56 | 77.3  | 88.0  | 56.0  | 74.0  | 0.66 | 52.6  | 50.9  | 29.6  | 62.2  | 0.31 |
> | Logistic Regression |   65.2   | 67.3  |  1.8  |  0.5  | 0.47 | 61.4  | 82.9  | 35.6  | 71.9  | 0.61 | 60.4  | 45.2  |  0.0  |  0.0  | 0.25 |
> |         MLP         |   78.1   | 71.6  | 51.5  | 68.1  | 0.57 | 85.0  | 90.6  | 70.5  | 85.7  | 0.71 | 60.4  | 39.1  |  0.0  |  0.0  | 0.26 |
> |         SVM         |   56.7   | 57.3  | 18.9  | 53.0  | 0.33 | 61.6  | 88.7  | 25.1  | 67.3  | 0.53 | 51.4  | 61.3  | 15.2  | 57.3  | 0.33 |
> |         KNN         |   80.4   | 73.8  | 55.3  | 68.6  | 0.52 | 87.4  | 86.8  | 74.8  | 87.3  | 0.75 | 57.4  | 49.6  | 10.8  | 45.3  | 0.12 |
> |         Vexi        |   68.1   | 55.5  | 41.7  | 66.9  | 0.42 | 65.3  | 62.2  | 33.4  | 72.9  | 0.29 | 54.9  | 41.3  |  8.5  | 47.7  | 0.10 |
> |      VaxiJen2.0     |   75.7   | 62.2  | 54.6  | 72.1  | 0.57 | 82.0  | 74.8  | 66.6  | 84.2  | 0.64 |   -   |   -   |   -   |   -   |   -  |
> |      VaxiJen3.0     |   83.3   | 58.7  | 63.2  | 75.1  | 0.63 | 68.1  | 62.5  | 42.3  | 75.4  | 0.35 | 39.0  | 35.7  |  0.0  | 55.9  | 0.00 |
> |      VirusImmu      |     -    |   -   |   -   |   -   |   -  | 58.8  | 59.5  | 18.1  | 63.6  | 0.17 |   -   |   -   |   -   |   -   |   -  |
> |   ProVaccine-Ankh   |   82.3   | 78.5  | 60.3  | 73.3  | 0.64 | 92.2  | 99.7  | 84.3  | 92.3  | 0.85 | 76.9  | 73.5  | 55.0  | 73.5  | 0.61 |
> |   ProVaccine-ESM2   |   80.6   | 77.0  | 57.0  | 71.7  | 0.66 | 90.3  | 99.5  | 80.6  | 90.6  | 0.82 | 74.0  | 61.7  | 46.1  | 68.2  | 0.58 |
> | ProVaccine-ProtBert |   84.5   | 84.5  | 65.9  | 77.0  | 0.66 | 91.4  | 97.4  | 82.8  | 91.2  | 0.84 | 71.5  | 68.7  | 44.7  | 67.6  | 0.54 |
>
>
> ProVaccine continues to demonstrate significantly better performance compared to other baselines. The new results have been updated in the revised manuscript.

---

> ### Author Response · Authors · 2024-11-22
> **Response (3)**
>
> 2. **Folding Model Dependencies**: We followed the suggestion and assessed the impact of structure quality on model performance by two additional experiments. We extracted AlphaFold2-predicted structures for all proteins in the constructed Immuno dataset and tested their performance. We retrieved pre-folded protein structures with UniProt IDs from the AFDB and used AlphaFold2 to predict structures for the remaining sequences without UniProt IDs. The test results are as follows:
>
> **Bacteria**:
> | Fold Type |        Model        | Accuracy |  TopK  |  MCC   |  F1    | KS-statistic |
> |:---------:|:-------------------:|:--------:|:------:|:------:|:------:|:------------:|
> |  ESMFold  |   ProVaccine-Ankh   |   82.7   |  78.8  |  61.8  |  75.0  |    0.632     |
> |           |   ProVaccine-ESM2   |   80.4   |  77.0  |  56.4  |  70.3  |    0.613     |
> |           | ProVaccine-ProtBert |   82.5   |  80.1  |  61.2  |  74.1  |    0.657     |
> | AlphaFold |   ProVaccine-Ankh   |   82.3   |  78.5  |  60.3  |  73.3  |    0.645     |
> |           |   ProVaccine-ESM2   |   80.7   |  77.0  |  57.0  |  71.7  |    0.655     |
> |           | ProVaccine-ProtBert |   84.5   |  84.5  |  65.9  |  77.0  |    0.659     |
>
> **Virus**:
> | Fold Type |        Model        | Accuracy |  TopK  |  MCC   |  F1    | KS-statistic |
> |:---------:|:-------------------:|:--------:|:------:|:------:|:------:|:------------:|
> |  ESMFold  |   ProVaccine-Ankh   |   91.5   |  97.5  |  83.0  |  91.4  |    0.845     |
> |           |   ProVaccine-ESM2   |   89.0   |  96.2  |  78.2  |  89.2  |    0.795     |
> |           | ProVaccine-ProtBert |   89.7   |  98.7  |  79.6  |  89.4  |    0.831     |
> | AlphaFold |   ProVaccine-Ankh   |   92.2   |  99.7  |  84.3  |  92.3  |    0.847     |
> |           |   ProVaccine-ESM2   |   90.3   |  99.5  |  80.6  |  90.6  |    0.822     |
> |           | ProVaccine-ProtBert |   91.4   |  97.4  |  82.8  |  91.2  |    0.844     |
>
> **Tumor**:
> | Fold Type |        Model        | Accuracy |  TopK  |  MCC   |  F1    | KS-statistic |
> |:---------:|:-------------------:|:--------:|:------:|:------:|:------:|:------------:|
> |  ESMFold  |   ProVaccine-Ankh   |   77.7   |  77.4  |  55.8  |  74.5  |    0.616     |
> |           |   ProVaccine-ESM2   |   73.7   |  67.8  |  45.5  |  67.1  |    0.582     |
> |           | ProVaccine-ProtBert |   70.3   |  67.8  |  41.6  |  67.1  |    0.510     |
> | AlphaFold |   ProVaccine-Ankh   |   76.9   |  73.5  |  55.0  |  73.6  |    0.611     |
> |           |   ProVaccine-ESM2   |   73.9   |  61.7  |  46.1  |  68.2  |    0.582     |
> |           | ProVaccine-ProtBert |   71.5   |  68.7  |  44.8  |  67.6  |    0.540     |
>
>
>
> For the second experiment, we tried to evaluate the prediction performance on a subset of the test dataset derived from the PDB database, which provides publicly available crystal structures. However, this subset consists of only 10 entries, which means the majority of the proteins have no crystal structure. Thus, this experiment could not be performed.
> We have added the result analysis and discussions in manuscript section 5.2.1.

---

> ### Author Response · Authors · 2024-11-22
> **Response (4)**
>
> **Questions**:
>
> 1. **homology cut-off**: In the initial submission, we only checked the similarity between positive and negative samples during dataset construction. In the revision, we followed your suggestion and applied an additional redundancy removal step between all training and test samples with a 90% similarity cut-off to avoid potential data leakage. We test the performance on Immuno-Bateria and use ESMFold2 for structure prediction due to the limited time of implementation. After this modification, we observed a reasonable decrease in performance across all models, including the baseline models. Despite this challenge, ProVaccine continues to outperform other models and demonstrates its robustness and effectiveness.
>
> **Bateria**
>  |        Model        | Accuracy |  TopK  |   MCC  |   F1   | KS-statistic |
> |:-------------------:|:--------:|:------:|:------:|:------:|:------------:|
> | Random Forest                              | 89.6     | 100.0 | 79.2  | 90.2  | 0.847        |
> | Gradient Boosting                          | 87.8     | 100.0 | 75.5  | 88.6  | 0.802        |
> | XGBoost                                    | 92.7     | 100.0 | 85.4  | 93.2  | 0.890        |
> | SGD                                        | 82.0     | 92.1  | 65.4  | 81.3  | 0.714        |
> | Logistic Regression                        | 62.1     | 90.5  | 34.0  | 73.5  | 0.661        |
> | MLP                                        | 85.4     | 91.2  | 70.6  | 86.1  | 0.738        |
> | SVM                                        | 78.4     | 92.9  | 57.8  | 77.9  | 0.614        |
> | KNN                                        | 90.1     | 93.1  | 80.3  | 90.5  | 0.803        |
> | Vexi                                       | 65.4     | 62.2  | 33.4  | 72.9  | 0.285        |
> | VaxiJen2.0                                 | 82.0     | 74.8  | 66.6  | 84.2  | 0.641        |
> | VaxiJen3.0                                 | 68.2     | 62.5  | 42.4  | 75.4  | 0.349        |
> | VirusImmu                                  | 58.8     | 59.6  | 18.1  | 63.6  | 0.175        |
> | ProVaccine-Ankh | 91.6     | 100.0 | 84.0  | 91.7  | 0.906        |
> | ProVaccine-ESM2 | 92.4 | 100.0 | 85.0  | 92.8  | 0.876        |
> | ProVaccine-ProtBert    | 94.9     | 100.0 | 89.8  | 95.0  | 0.920        |
>
> 2. **Figure Caption**: Thank you for the suggestion. We did not provide more details in the initial submission due to the page limit. In the revision, we have updated the manuscript to include more informative captions.
>
> 3. **Case Study 1**: Thank you for your suggestions on the post-hoc analysis. In the revised manuscript, we modified the description and provide formulation of recall and enrichment to improve clarity and understanding.
>
> 4. **Interpretatability**: We completely agree with your perspective that model interpretability is a crucial consideration, especially in addressing scientific problems like immunogenicity prediction. This was indeed one major motivation for incorporating the attention pooling layer and designing the third experiment in Section 5.3.2. Our experimental design aligns closely with the idea you mentioned here. We assume that for a given protein, each AA has an importance score indicating whether it is immunogenic. In our implementation, this importance score is quantified using the attention scores from the pooling layer.
> Although this implementation is still relatively simple, it already demonstrates that our model is capable of identifying some critical regions as plausible epitopes, as illustrated in the example of the spike protein visualized in Figure 5. We believe this exploration holds promise for future applications to other deep learning-based methods. Furthermore, it presents a meaningful and worthwhile direction for advancing epitope recognition using deep learning approaches.
>
> 5. **Generalizability to Other Pathogens**: We believe the results from both post-hoc analyses address the generalizability of our model to other pathogens. In both cases, we tested and analyzed the pretrained model on entirely new proteins. In other words, **the 1,858 candidates for Helicobacter pylori examined in Section 5.3.1 and the 38 candidates for SARS-CoV-2 tested in Section 5.3.2 were not included in ProVaccine's current training dataset**.
>
> Thank you once again for all the suggestions. We believe that the revised manuscript has significantly improved in quality and is now qualified for publication.

---

> > ### Comment · Reviewer_G2xM · 2024-11-26
> >
> > I would like to thank the authors for their detailed response and addressing majority of my comments.

---

> > > ### Author Response · Authors · 2024-11-29
> > >
> > > Thank you again for your encouraging assessment and for your constructive feedback.

---

### Official Review · Reviewer_Xkx7 · 2024-11-05

**Soundness:** 3
**Presentation:** 3
**Contribution:** 3
**Rating:** 8
**Confidence:** 3

**Summary:**

The authors present the ProVaccine model for immunogenicity prediction, as well as the Immuno datasets of antigens from bacteria, viruses and tumors, which is balanced across protective and nonprotective antigens. They show that ProVaccine is more accurate compared with both baseline ML and hosted models.

**Strengths:**

- novel model architecture
- contribution of the Immuno datasets
- ProVaccine achieves mostly top performance as reported in Table 1

**Weaknesses:**

- It's not obvious what the contribution of any of the representations (sequence, fine, coarse, and descriptors) is to model performance. What if only the sequence representation is used as input? Doing ablation studies is essential for understanding these contributions, and should be included in this work.
- Figure 4 is hard to interpret, it would be easier to understand if instead the likelihood stats of the 11 determined immunogens under each model were reported.
- The appendix is missing:
  - page 7 in section 5.2.1 refers to Tables 4-6 in the appendix
  - page 8 in section 5.3.1 refers to recall and fold-enrichment metrics that do not appear in any table. Also reference is made to Table 10, which does not appear to exist.
  - page 9 in section 5.3.2 refers to Table 11 in the appendix, which does not exist

**Questions:**

- I don't understand the approach to measuring accuracy described in the first paragraph of section 5.2.1. Are you doing hyperparameter optimization on the validation set for each split? Why randomly select "50% of the test data to calculate the scores for each metric" since you already have a validation set in your split?
- It's not clear what the generalizability of section 5.2.2 is measuring. Should a immunogenicity predictor generalize across bacteria, viruses and humans?
- For the three conclusions about ProVaccine in the final paragraph of subsection 5.3.1, I don't understand how the authors concluded the second and third - could you please explain these in greater detail?

---

> ### Author Response · Authors · 2024-11-22
>
> Thank you for your review and for recognizing the novelty of our work in terms of algorithmic construction, dataset preparation, empirical performance, and so on. We noticed that the concerns raised mainly stem from unfound tables and figures, which **we have provided in the appendix and uploaded separately as supplementary material**. Below, we address each of the mentioned weaknesses and questions in detail.
>
> **Weaknesses**:
>
> 1. **Ablation Study**: The required ablation studies on different modules have been provided in Tables 7–9 in the appendix (uploaded as supplementary material). These studies include experiments with the following inputs: PLM-encoded sequence only, PLM-encoded sequence + atomic tokenization, PLM-encoded sequence + physiochemical descriptors, and the complete version combining PLM-encoded sequence (finer-level information) + atomic tokenization (coarser-level information) + physiochemical descriptors. Furthermore, following your suggestion, we conducted an additional experiment that directly predicted the information from one-hot encoded sequences. The results have been updated in the same tables in the revised appendix.
> In summary, encoding sequence information with PLMs is essential for enhanced performance in immunogenicity prediction tasks. Additionally, both atomic tokenization and physiochemical descriptors contribute significantly to the representation of the protein. The combination of all three modules achieves optimal prediction performance.
>
>
> 2. **Likelihood values in Figure 4**: Thank you for suggesting a clearer way of reading Figure 4. We have now included a detailed likelihood report for the 11 determined immunogens under each model in Table 15 of the revised Appendix. We believe this additional information provides clearer insights into the performance of different models and their ability to identify known protective antigens.
>
> 3. **Missing Tables and Figures**: The appendix has been uploaded separately as the supplementary material, which includes all the mentioned reference Tables and Figures in the comment (Tables 4-6, 10, and 11). Note that Table 11 in the initial submission is now Table 12.
>
>
>
> **Qestions**:
>
> 1. **Random Splits on the Test Dataset**:We repeated the construction of random test sets to obtain the variance in the model’s predictive performance. This repetitive evaluation process is conducted entirely on the test set and is independent of validation or hyperparameter tuning.
> As explained at the beginning of Section 5.2.1, our data is split into training, validation, and test sets in a 7:1:2 ratio. We use 70% of the data as the training set to train the model and compute accuracy on the 10% validation set for hyperparameter optimization. For the 20% test set, we performed 10 random sampling rounds with each round picking up 50% of the test sets as a new “sub-test set” and calculated the scores of the trained model on these sub-test sets.
>
> 2. **Generalizability Evaluation**: In Section 5.2.2, we designed experiments to evaluate the generalizability of the model when the training and test sets come from different species (lines 359-360). For instance, the orange bar in the top left corner of Figure 3 represents a model trained on virus data and tested on bacteria data. Contrary to the question raised in the review, we did not train the model on additional datasets and then test it separately on the virus, bacteria, and tumor datasets. The reason to design such an experiment is that immunogenicity data varies significantly in abundance across different species, making it potentially infeasible to retrain a model specifically for any given species. Therefore, we desire a well-designed model that could generalize and perform well on test data from species different from the training data.
>
> 3. **Interpretation of Case Study 1**: For the two points in the case study, (2) The majority of antigens in nature are non-immunogenic, meaning they should be assigned a negative label in our prediction task. Therefore, an ideal predictive model should assign probabilities closer to 0 for most data points, rather than distributing them more uniformly across the 0-1 range (as observed in many other baseline methods) or concentrating them in a very narrow region (as seen in logistic regression). (3) The time and financial costs of biological experiments are significant. Therefore, in addition to capturing true positives, it is preferable for a model to minimize false positive predictions. As shown in Table 10 of the Appendix, our method assigns the fewest positive labels (i.e., the fewest PVCs) while successfully identifying all 11 true positives. This demonstrates that our method can identify positive immunogens at a lower experimental cost.
>
> We believe the above responses clearly address your concerns and questions. We look forward to your feedback and are happy to provide further clarification on any remaining issues.

---

> ### Comment · Reviewer_Xkx7 · 2024-11-25
>
> Thank you for your response, and my apologies for not having access to the supplementary material while reviewing, I don't know why I did not.
>
> * **Ablation studies**: Thank you for these. It is interesting that just using the fine and coarse sequence models do better on most tasks, with Recall being an exception (NB: I wish the same color scheme used in Tables 4-6 to indicate top performers would have been used here to make this clearer). Why then also add the physchem descriptors to ProVaccine?
> * **Likelihoods**: Thank you, this is much easier to interpret.
> * **Splits of test dataset**: doing what you've done says more about the distribution of the test set, which is apparent in the similar scales of stdevs reported, than it does about the model, e.g., the effect of random initialization or training on different subsets.
> * **Generalizability evaluation**: Apologies for not being clearer, but I was asking the specific question of why we should expect the model trained on one species to generalize to another. Is it clear that the immunogenic responses should be the same across species?
> * **Interpretation of Case Study 1**: Thank you, this sufficiently answers my questions.
>
> I've raised my score, looking forward to discussing further.

---

> ### Author Response · Authors · 2024-11-26
>
> Thank you for your continued engagement and for raising your score. We appreciate your thorough review and address your points below.
>
> 1. **Ablation Studies**:
>
> Following your suggestion, we have updated the tables in the appendix to highlight the best-performing results. Regarding the inclusion of physiochemical descriptors, according to the results of the ablation studies reported in Tables 12–14, using a combination of sequence, structure, and physiochemical descriptors (row 4) generally outperforms using only sequence features (row 1) or sequence + structure features (row 2) within the same PLM encoder. This observation aligns with other machine learning-based immunogenicity prediction studies, which typically rely on these physiochemical descriptors for predictions. It also suggests that while deep learning-based sequence and structure embeddings can be powerful in general representation learning tasks, molecular patterns derived from prior domain knowledge can still provide valuable additional information for specific downstream tasks. For instance, in the Table 12, when use Ankh to find the sequence embedding, we have:
>
> | Input Module | AUC-ROC | Accuracy | Precision | Recall | F1 | MCC           | TopK-30       | Cross Entropy | KS Statistic   |
> |:-|:-:|:-:|:-:|:-:|:-:|:-:|:--:|:-:|:--:|
> |**row 1** Sequence (PLM)           | 86.7±1.7      | 81.5±1.5      | 78.0±2.8      | 70.7±2.5      | 74.2±2.3      | 60.0±3.3      | 63.1±3.2      | **0.42±0.02** | 0.84±0.04      |
> |**row 2** Sequence (PLM) + Structure Token | 86.7±1.9  | 83.1±1.5      | 82.2±2.2      | 70.2±4.1      | 75.7±2.7      | 63.4±3.3      | 63.1±3.6      | 0.45±0.04     | **0.85±0.03**  |
> |**row 3** Sequence (PLM) + Descriptors | 87.1±1.7      | 82.2±1.6      | 77.1±3.4      | 72.4±3.5      | 74.6±2.7      | 61.1±3.6      | 63.4±3.9      | 0.42±0.03     | 0.81±0.05      |
> |**row 4** ProVaccine        | **89.8±1.7** | **86.2±0.8** | **84.5±1.3** | **77.7±2.3** | **80.9±1.5** | **70.3±1.7** | **88.7±4.0** | 0.44±0.04 | 0.71±0.03 |
>
>
> 2. **Splits of Test Dataset**:
>
> We agree with you that the random split of the test set primarily reflects the distribution of the data, which was our primary motivation for designing this test, i.e., to evaluate the stability of the trained model's performance on data with inherent variance.
>
> We also recognize the importance of examining model stability and have therefore conducted additional experiments by repeatedly training and testing the model on randomly split datasets. We conduct the evaluation on the Immuno-Bacteria dataset with an 8:1:1 random split over 5 repetitions. As shown in the table below, our model demonstrates considerable stability, maintaining consistent performance across different dataset partitions, indicating its resilience to random initialization and variations in training subsets. The results have also been added to Table 8 of the revised Appendix.
>
> | Model | AUC-ROC | Accuracy | Precision | Recall | F1 | MCC | TopK-30 | Cross Entropy | KS statistic   |
> |:-|:-:|:-:|:-:|:-:|:-:|:-:|:--:|:-:|:--:|
> | ProVaccine-Ankh      | 87.3±2.2      | 81.3±1.9      | 80.1±5.6      | 70.8±8.2      | 74.7±2.8      | 60.7±3.0      | 84.9±1.8      | 0.85±0.24     | 0.63±0.03      |
> | ProVaccine-ESM2      | 87.4±1.5      | 80.3±1.3      | 77.7±3.4      | 69.8±3.6      | 73.4±2.2      | 58.1±2.9      | 81.4±3.4      | 0.63±0.30     | 0.62±0.04      |
> | ProVaccine-ProtBert  | 88.4±1.8      | 81.9±1.8      | 77.8±2.6      | 74.8±6.2      | 76.1±2.4      | 61.7±3.9      | 83.8±3.2      | 0.50±0.10     | 0.66±0.05      |
>
>
> 3. **Generalizability Evaluation**:
>
> Before giving a detailed justification, we would like to first clarify that our model predicts immunogenicity specifically in humans, regardless of the species origin of the antigens. Thus, we evaluate the generalizability from the perspective of **whether a model trained on one species' antigens can predict the immunogenicity of antigens from another species, in the context of human immune responses**.
>
> We understand that there are differences in immunogenic responses across species, but our focus is on the shared immunogenic properties that antigens from different species may exhibit when interacting with the human immune system. Proteins from various species can elicit immune responses in humans, suggesting that there are commonalities that can be leveraged for prediction, despite their different origins. In this regard, we desire a model to capture these shared immunogenic features, such as when the training data comes from one species and the test data from another (in the context of human immunogenicity). This approach allows us to evaluate the model's performance and its capacity to generalize across different protein sources while maintaining the focus on human immune responses.
>
> We hope this round of discussion provides solid evidence or justification for the previously unresolved questions, and we look forward to your feedback.

---

> > ### Comment · Reviewer_Xkx7 · 2024-11-27
> >
> > Thank you to the authors for updating tables, performing additional experiments, and addressing questions. I have again raised my score.

---

> > > ### Author Response · Authors · 2024-11-29
> > >
> > > Thank you for your thoughtful review and the updated score. We’re grateful for your time and constructive insights!

---

### Meta-Review · Area_Chair_d6s1 · 2024-12-17

**Metareview:**

The paper presents ProVaccine, a deep learning model for predicting immunogenicity using a dual-attention mechanism and a comprehensive dataset of over 9,500 antigens. It demonstrates superior performance compared to traditional methods, particularly in identifying potential vaccine candidates.

Strengths include the novel model architecture, the extensive Immuno dataset, and comprehensive benchmarking against existing methods. The dual-attention mechanism effectively integrates multimodal data, enhancing prediction accuracy.

Weaknesses involve potential biases in the negative dataset construction.

The decision to accept is based on the innovative approach to immunogenicity prediction and the potential impact of the comprehensive dataset.

**Additional Comments On Reviewer Discussion:**

Reviewer Xkx7 raised problems on ablation studies. Reviewer G2xM and Reviewer BQLB asked questions on data construction and data splitting. The authors provided more experiments, which basicly answered those questions.

---

### Decision · Program_Chairs · 2025-01-22

Accept (Poster)